# Mesenchymal Stem Cells and Begacestat Mitigate Amyloid-β 25–35-Induced Cognitive Decline in Rat Dams and Hippocampal Deteriorations in Offspring

**DOI:** 10.3390/biology12070905

**Published:** 2023-06-25

**Authors:** Asmaa Gaber, Osama M. Ahmed, Yasser A. Khadrawy, Khairy M. A. Zoheir, Rasha E. Abo-ELeneen, Mohamed A. Alblihed, Ahlam M. Elbakry

**Affiliations:** 1Comparative Anatomy and Embryology Division, Department of Zoology, Faculty of Science, Beni-Suef University, Beni-Suef P.O. Box 62521, Egypt; 2Physiology Division, Department of Zoology, Faculty of Science, Beni-Suef University, Beni-Suef P.O. Box 62521, Egypt; 3Medical Physiology Department, National Research Center, Giza P.O. Box 12622, Egypt; 4Cell Biology Department, National Research Center, Giza P.O. Box 12622, Egypt; 5Department of Medical Microbiology, college of medicine, Taif University, P.O. Box 11099, Taif 21944, Saudi Arabia

**Keywords:** Aβ, MSCs, γ-secretase inhibitor, microglial cells, brain-derived neurotrophic factor, proinflammatory cytokines

## Abstract

**Simple Summary:**

Alzheimer’s is a type of dementia that affects memory, thinking, and behavior. It is a chronic neurological illness. Alzheimer’s disease is the primary cause of dementia in senior individuals and the sixth greatest cause of death in the world. Although several medications are used to treat Alzheimer’s disease, none of them actually prevent the illness from progressing. The groundbreaking finding of stem cells has given rise to fresh optimism for the creation of Alzheimer’s disease-modifying therapies. The therapeutic use of secretase inhibitor (Begacestat) for the treatment of Alzheimer’s disease significantly lowers the amounts of amyloid proteins. The current study focuses on how secretase inhibitor and mesenchymal stem cells are used to treat Alzheimer’s disease in pregnant female rats and how this affects the development of the progeny. According to current research, mesenchymal stem cells or γ-secretase inhibitors treatment against single amyloid protein 25–35 injection for dams repaired histopathological changes, inhibited microglial cell activity, improved behavioral impairments, reduced neuroinflammatory cytokines, and decreased the protein concentration of p-tau and the amyloid precursor protein by increasing the activity of the brain-derived neurotrophic factor, decreasing the expression of NF-κB. Therefore, this study proved a possible protection against Alzheimer’s disease by mesenchymal stem cells and γ-secretase inhibitor.

**Abstract:**

Alzheimer’s disease (AD) is the most common cause of age-related neurodegeneration and cognitive decline. AD more commonly occurs in females than in males, so it is necessary to consider new treatments specifically targeting this population. The present study investigated the protective effects of Begacestat (γ-secretase inhibitor-953, GSI-953) and bone marrow-derived mesenchymal stem cells (BM-MSCs) during pregnancy on cognitive impairment in rat dams and neurodegeneration in offspring caused by the intracerebroventricular injection of Aβ 25–35 before pregnancy. The performances of dams injected with amyloid-β 25–35 (Aβ 25–35) during behavioral tests were significantly impaired. The offspring of Aβ 25–35-injected dams treated with BM-MSCs or GSI-953 showed a dramatically reduced number and size of activated microglial cells, enhancement in the processes length, and a decrease in the proinflammatory cytokine levels. Additionally, BM-MSC or GSI-953 therapy reduced Aβ 25–35-induced increases in tau phosphorylation and amyloid precursor protein levels in the neonates’ hippocampus and elevated the lower levels of glycogen synthase kinase-3 and brain-derived neurotrophic factor; moreover, reversed Aβ 25–35-induced alterations in gene expression in the neonatal hippocampus. Finally, the treatments with BM-MSC or GSI-953 are globally beneficial against Aβ 25–35-induced brain alterations, particularly by suppressing neural inflammation, inhibiting microglial cell activation, restoring developmental plasticity, and increasing neurotrophic signaling.

## 1. Introduction

Alzheimer’s disease (AD) is an untreatable neurodegenerative disease and one of the most common causes of dementia as it accounts for 60–80% of global dementia cases [1]. Worldwide, AD affects about 50 million individuals, and epidemiological models project that there will be 152 million cases by 2050, i.e., roughly a six-fold rise in the number of AD cases with that in 2006 [2]. Furthermore, roughly two-thirds of Alzheimer’s patients in the United States are women, according to the Alzheimer’s Association [3]. The pathology of AD is characterized by progressive neuronal cell death in a large region of the cerebral cortex, basal forebrain, and hippocampus, leading to emotional and cognitive impairments such as memory loss, impaired decision making, and language difficulties [4,5,6,7]. The pathological hallmarks of AD are the formation of intracellular neurofibrillary tangles (NFTs) containing hyperphosphorylated tau protein, neuroinflammation, extracellular deposition as plaques of amyloid-β peptide (Aβ), and other alterations in synaptic structure [7,8,9]. Numerous studies have suggested that AD pathogenesis entails several neurodegenerative processes induced by plaque buildup and that different AD risk genes impact all these pathogenic mechanisms [10]. One of the most significant degenerative processes is the production of reactive oxygen species by amyloid plaques, inducing oxidative cell damage and activating inflammatory cascades [11].

Numerous treatments for AD have been proposed, but the majority simply addresses the symptoms and none are able to halt or reverse the course of the illness [12]. In addition, the effectiveness of AD medications currently on the market barely reaches 20% [13]. Given the significance of amyloid plaque deposition and NFTs formation in the early stages of AD, pharmacological interventions that can lower the Aβ load, prevent tau phosphorylation, or shield sensitive neurons from subsequent pathogenic processes, may stop the resulting cognitive impairments [14].

The enzymatic cleavage of transmembrane amyloid precursor protein (APP) by β- and γ-secretases results in the formation of amyloid plaques [15]. Because γ-secretase is a multisubunit complex that catalyzes the last step of Aβ production, inhibiting this enzyme may stop the aggregation of amyloid peptide (Aβ) [16,17]. Begacestat (or γ-secretase inhibitor-953, GSI-953) is a γ-secretase inhibitor that lowers Aβ plasma levels in a phase I clinical trial. Cellular tests using Notch showed that Begacestat is 15 times more effective in inhibiting APP cleavage [18]. The same authors noted that, while modest doses of GSI-953 still reduced Aβ1-40 levels in the brain and plasma, large doses significantly decreased the formation of Aβ1–40 in the brain, cerebrospinal fluid (CSF), and plasma [18]. Importantly, Begacestat has been shown to improve the contextual memory deficits of Tg2576 transgenic mice, a model of AD [18], indicating Begacestat potential as a therapy for AD. However, more research is necessary.

In addition to small-molecule medicines that block the pathogenic processes leading to AD, stem cell therapy is becoming more popular as a neuronal replacement and/or protective measure [19,20,21]. In fact, stem cell treatment in animal models of neurodegenerative diseases has demonstrated encouraging outcomes [22,23]. The most popular stem cell types used in research on therapeutic applications are pluripotent stem cells (iPSCs), brain-derived neural stem cells, and bone marrow-derived mesenchymal stem cells (BM-MSCs) [24,25] because these cells can differentiate into a variety of functional cells, including neural cells [26]. Stem cell-based therapy may be more effective against AD than conventional drug therapies because implanted stem cells can improve the brain microenvironment by supplying growth-promoting and growth-permissive factors for synaptogenesis and neurite repair, reduce oxidative stress by enhancing local antioxidant capacity, and stimulate the sustained production of neurotrophic factors such as brain-derived neurotrophic factor (BDNF) and nerve growth factor (NGF) [27].

Early AD treatment offers a chance to stop or decrease behavioral dysfunction and synaptic malfunction, which are pathogenic processes known to begin long before clinical diagnosis [28]. Therefore, the present study assessed the effectiveness of BM-MSCs and GSI-953 in reducing cognitive impairments induced by Aβ 25–35, which is the toxic Aβ fragment, in adult rat females and neurodegeneration in these females’ offspring.

## 2. Materials and Methods

### 2.1. Experimental Animals

In the current study, 80 Wistar rats, namely 50 mature females and 30 mature males (Table 1) for mating, and weighing 200–250 g, were employed. The animals were purchased from the Egyptian Organization for Biology Products and Vaccines (VACERA) animal housing facility in Egypt. Animals were housed under close observation in well-ventilated stainless steel cages at the Department of Zoology animal house. They were kept under a regular daily dark/light cycle, controlled humidity (50% ± 5%), and controlled air temperature (25 °C ± 5 °C) to prevent infections. Animals had access to standard tap water and were fed a standard rat pellet diet along with some vegetables as a source of vitamins. The Beni-Suef College Animal Care and Use Committee’s broad guidelines were followed for the animal care procedure (Approval number 019-75).

### 2.2. Surgical Procedure

Female rats underwent surgery as described previously [29]. Free Aβ 25–35 was dissolved to a concentration of 1 mg/mL in 0.9% saline and was aggregated by in vitro incubation at 37 °C for 4 days [29], to obtain a solution of Aβ 25–35 oligomers (oAβ 25–35). For profound anesthesia, 7 mg/kg xylazine and 70 mg/kg ketamine were injected intraperitoneally into rats. Head hair was then shaved using surgical shears. After a betadine sanitizing treatment, a biodegradable surgical towel was used to cover the shaved region. A longitudinal incision along the median longitudinal calvaria was performed to gently separate the subcutaneous tissue and fascia. Sterile dry cotton was employed to halt the bleeding, and bregma was marked with a pen. A 1 mm diameter hole was drilled into the bone 0.8 mm posterior to the bregma and 2.0 mm lateral to the midline (above the lateral ventricle) using a flexible bone drill. The bleeding was stopped, and sterile cotton was used repeatedly to clean the surface of the skull. A needle connected to a Hamilton microsyringe was gently pushed through the borehole at a depth of 4.6 mm and used to progressively inject 10 µL Aβ 25–35 (10 ug). The needle was left in place for two min to allow for the complete draining of the syringe content into the lateral ventricle. An identical volume of 0.9% saline was injected into control rats. Betadine was used to disinfect the wound before using a quick stitch technique to close the wound. Rats were subjected to the Y-maze and novel object recognition tests after 10 days of surgery to assess their working and reference memories, respectively. The female rats that underwent surgery as described previously were mated for one or two days with a male during the proestrus stage. The first day of pregnancy was determined by looking for sperm in a vaginal smear or vaginal plug. Number of successful pregnant animals was 7–9 rats for each group and each pregnant animal birth at arrange of 6–9 pups.

### 2.3. Animal Grouping: (Figure 1)

Female Wistar rats were divided into five groups. Before pregnancy, rats of group G1 (*n* = 10) were intracerebroventricularly (i.c.v.) injected with 10 µL saline (0.9%) and rats of group G2 (*n* = 30) were i.c.v. injected with 10 µL Aβ 25–35/rat. During pregnancy, rats of group G3 (*n* = 10) were intravenously (i.v.) injected with Dulbecco’s modified Eagle medium (DMEM) (4.5%). After ten days of i.c.v. Aβ 25–35 injections and starting of pregnancy, adult female rats of group 2 were divided into Group G4 (*n* = 10) and were intravenously injected with BM-MSCs once a week for three weeks during pregnancy and group G5 were orally given 2.5 mg/kg body weight Begacestat during pregnancy every other day; the remaining 10 animals of group 2 acted as an AD group (negative control). Blood samples were left to clot and were then centrifuged at 3000 rpm for 15 min to obtain serum and hippocampus of offspring were removed on postnatal days (PND) 7, 14, and 21 for histological, immunohistochemical, and neurochemical analyses.
Figure 1Design of the study, adult female albino rats were subjected to intracerebroventricular injection (i.c.v.) with Aβ 25–35 and vehicle (0.9% saline) before pregnancy and allowed to recover for 10 days before being subjected to behavioral tests (novel object recognition test and Y-maze test). After that, the female rats were mated with males, and the pregnant female rats subjected to Aβ 25–35 injection were classified into AD group, AD group received intravenous injection of MSCs (one million/rat/week), and the AD group orally treated with GSI-953 (2.5 mg/kg b.wt). The neonates were sacrificed on different postnatal days (7, 14, and 21) and brain hippocampus was subjected to histological, immunohistochemistry, qRT–PCR, and Western blotting study.
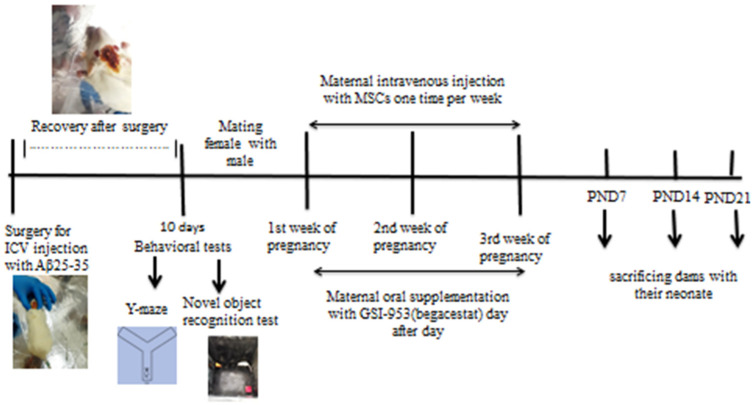


### 2.4. Isolation of BM-MSCs from Rats: (Figure 2)

Mesenchymal stem cells (MSCs) were extracted from rat bone marrow as described in previous work [30,31,32]. The bone marrow was flushed out of the femur using 4.5% DMEM (Life Science Group Ltd., Sandy, UK), and the solution was centrifuged at 3000 rpm for five minutes. A complete medium containing 4.5% DMEM, 15% fetal bovine serum (Lonza Verviers Sprl, Verviers, Belgium), and 1% penicillin/streptomycin (Life Science Group Ltd., Sandy, UK) was used to seed isolated bone marrow cells in culture flasks. The cells were then maintained at 37 °C in a 5% CO_2_ environment with 50% humidity. Three to four days later, the medium was changed, and when the cells reached 80%–90% confluence (7–10 days after seeding) they were harvested by incubating them with 0.25% trypsin/1 mm EDTA (Greiner Bio-One, Stuttgart, Germany) for five minutes at 37 °C, centrifuged, and resuspended in DMEM. Viability was assessed before injection by mixing 10 µL of the cell suspension with 10 µL of 0.4% trypan blue solution and counting the number of stained (dead/dying) cells using a hemocytometer. Cells with a 95% viability rate were employed for injection. The isolated cells were previously characterized in our publication [33]. Approximately one million cells were injected each week into the animals for three weeks.
Figure 2Morphology of the isolated undifferentiated mesenchymal stem cells from bone marrow under inverted microscopy: (**a**) MSCs in the culture flask at first day of isolation showed rounded cells; (**b**) MSCs begin to convert into a flat fibroblast-like morphology after 7 days of isolation; and (**c**,**d**) MSCs at day 10 day of isolation before and after washing with fresh DMEM, respectively.
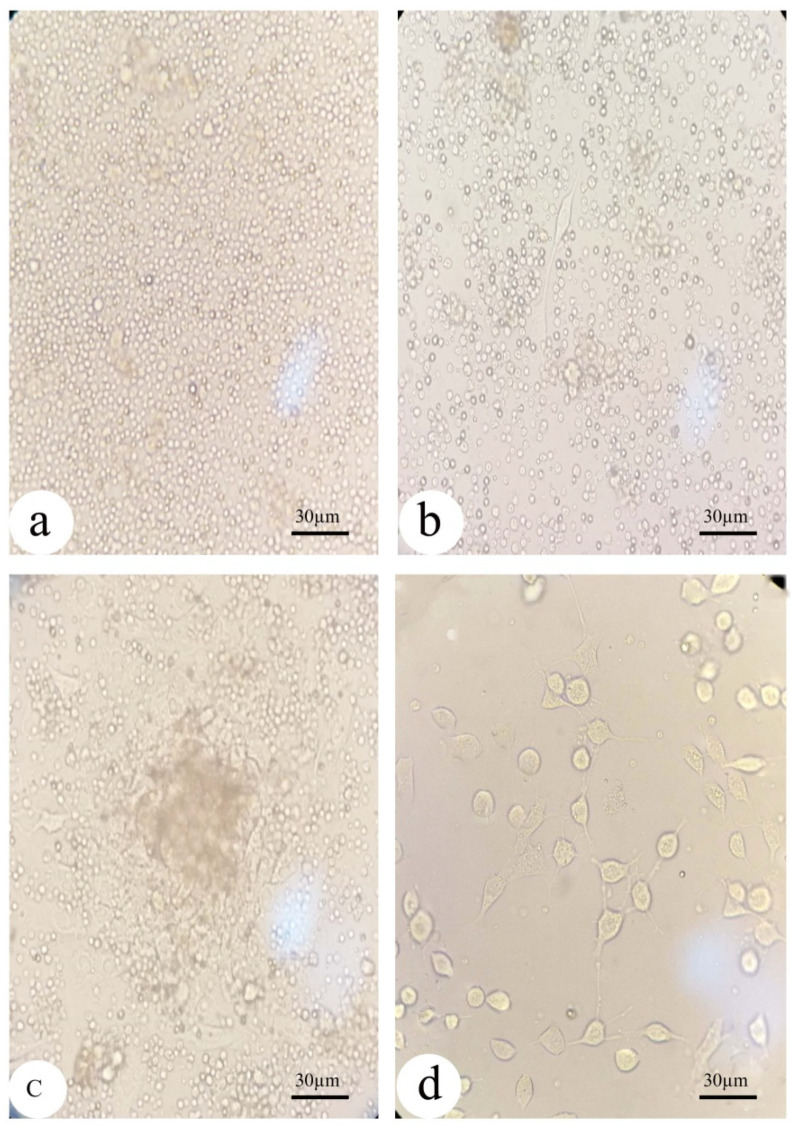


### 2.5. Behavioral Tests

#### 2.5.1. Y-Maze Test

The Y-maze task was employed to evaluate spatial working memory. The wooden Y-maze has three evenly spaced arms, each being 50 cm long, 5 cm broad at the hub, and 10 cm wide at the extremity, which is enclosed by 20 cm high walls. The third arm was blocked by a barrier, and individual rats were placed in one arm and permitted to freely explore that arm and the neighboring arm for eight minutes. Afterwards, the barrier was removed, and the rats were given 30 min to freely explore each arm. We kept track of the number of entries, the time spent in the two previously explored (familiar) arms, and the time spent in the previously blocked (new) arm. The working memory performance was measured as a percentage of time (%) spent in the novel arm compared with that spent in the familiar arms [34].

#### 2.5.2. Novel Object Recognition Test

A three-day strategy called the new object recognition test was used to assess long-term memory [35]. On the first day, all animals were habituated to a wooden cage of 30 × 30 × 30 cm for 10 min [35,36]. On day two, animals were given 10 minutes to freely explore the same cage, in which two identical new wooden objects had been placed at opposite corners, 2 cm from the walls [37]. On the third day, an object was exchanged for another one of different shape, size, and color, and the rats were given five minutes to investigate. The recognition index (RI) was determined by dividing the time spent analyzing the novel object by the sum of the times spent investigating both objects [35]. A higher RI is considered as an indicator of better recognition memory [38,39].

### 2.6. Histological Analysis of the Newborn Hippocampus

The whole hippocampus of neonates from each treatment group (four per group) was removed, kept in 4% paraformaldehyde for 48 h at 4 °C, dehydrated using a concentration gradient of ethanol, and then embedded in paraffin. Sections (5 µm) were generated using microtome stained with hematoxylin and eosin at room temperature for 12 min, and then observed under a light microscope to assess the histopathological alterations.

### 2.7. Congo Red Staining

The dehydrated and deparaffinized hippocampus sections were stained with Congo red for 20 min at room temperature, and the sections were differentiated using alkaline alcohol for seconds before they were rinsed in running water for 5 min. The sections were immersed in hematoxylin for 2 min and then rinsed in tap water until it turned blue. Finally, the sections were cleared in xylene and then covered with DPX.

### 2.8. Immunohistochemistry

Hippocampus paraffin sections were cut at a thickness of 5 µm, blocked for one hour in a solution containing 1% bovine serum albumin (BSA) and 0.3% Triton X-100, and then incubated overnight at 4 °C with an antibody recognizing the microglial marker ionized calcium-binding adapter molecule-1 (Iba-1) from Abcam (Cambridge, UK). Afterwards, sections were exposed for one hour to a corresponding secondary antibody horse radish peroxidase (HRP). The immunostaining was visualized under a 10× and 100× objective lens on an Olympus microscope. Image J was used to quantify the number, size, and dimensions of the microglial cells (NIH, Bethesda, MD, USA).

### 2.9. ELISA Assay

Alterations in inflammatory cytokines (IL-1β, TNFα), GSK-3β, and BDNF in the newborn cortex were detected by ELISA kits of IL-1β (SEA563Ra) (Cloud-Clone Crop., Katy, TX, USA), TNFα (Biolegend, San Diego, CA, USA), GSK-3β (Abbexa LLC, Houston, TX, USA), and BDNF (SEA011Ra) (Cloud-Clone Crop., Katy, TX, USA). All the ELISA kits were used in accordance with the manufacturer’s instructions, and the quantity of these factors was expressed as pg/mL.

### 2.10. qRT-PCR

Total RNA was extracted from hippocampus lysates (three samples for each group) using the Direct-zol RNA Miniprep Plus kit (cat. #R2072, Zymo Research, Irvine, CA, USA) according to the manufacturer’s instructions and reverse transcribed using the Superscript IV One-Step RT-PCR kit (cat. #12594100, Thermo Fisher Scientific, Waltham, MA, USA). The sequences of primer pairs specific from the target genes (BDNF, nuclear factor-κB (NF-κB), tumor necrosis factor receptor (TNFR), transforming growth factor-β (TGF-β), and caspase-3) are indicated in Table 2. Primers were used to amplify cDNA with the 2X Platinum TM SuperFiTM RT-PCR Master Mix. SYBER green (Step One, Applied Biosystem, Foster City, CA, USA) was used for qPCR, and the delta-delta Ct (∆Ct) method was used to calculate the ratio of the target gene expression level to that of the β-actin gene.

### 2.11. Western Blotting

Total proteins were extracted from hippocampus samples using a total protein extraction kit (Bio-Rad cat. #163-2086, Tokyo, Japan) following the manufacturer’s instructions. The Bradford Protein Assay Kit (SK3041) for quantitative protein analysis was provided by Bio basic Inc. (Markham, ON, Canada) and performed according to manufacturer’s instructions to determine the protein concentration in each sample. Equal amounts of proteins were separated using sodium dodecyl sulfate polyacrylamide gel electrophoresis (SDS-PAGE) according to their molecular weight and then transferred to polyvinylidene fluoride membranes. Polyacrylamide gels were performed using the TGX Stain-Free™ FastCast™ Acrylamide Kit (SDS-PAGE), which was provided by Bio-Rad Laboratories Inc. Cat #161−0181. The SDS-PAGE TGX Stain-Free Fast Cast was prepared according to the manufacturer’s instructions. The membrane was blocked in tris-buffered saline with Tween 20 (TBST) buffer and 3% bovine serum albumin (BSA) at room temperature for 1 h and then overnight at room temperature with primary antibodies recognizing tau phosphorylated Thr231 (1:1000), APP (1:1000), and β-actin (1:1000) (Cell Signaling Technology. Danvers, MA, USA). Following three to five TBST washes, the blots were incubated with horseradish peroxidase conjugated 2ry antibodies at a dilution of 1:5000 for 30 min and then exposed to a chemiluminescent substrate (Clarity^TM^ Western ECL substrate, Bio-Rad cat. #170-5060) to detect immunoreactive bands. The target band signals were visualized using a CCD camera-based imager (ChemiDoc MP imager), quantified, and normalized to β-actin signals using the accompanying image analysis programmer.

### 2.12. Statistical Analysis

All statistical analyses were conducted using SPSS version 17.0 (SPSS Inc., 1989–2007; Chicago, IL, USA). Data are presented as the mean ± standard error of the mean after one-way ANOVA analysis. The effects of age, therapy, and their interactions on most outcomes were examined by two-way ANOVA followed by Tukey’s multiple comparison tests. Behavioral tests results were evaluated by one-sample T-test. The significance was considered at three levels, namely at *p* < 0.05, *p* < 0.01, and *p* < 0.001, while *p* > 0.05 was non-significant.

## 3. Results

### 3.1. Aβ 25–35 i.c.v. Injection Impaired Working and Object Recognition Memories in Adult Female Rats

Rats i.c.v. injected with Aβ 25–35 (group G2) showed significant working memory impairments in the Y-maze test (Table 2) as the time spent in the new arm was decreased compared with that of rats receiving saline (group G1). Additionally, the lack of differences among groups in the number of novel arm entries suggested that there was no effect of the treatment on the motor activity or exploratory drive (*p* > 0.05). Moreover, the novel object recognition task revealed that G2 rats explored novel objects less than G1 rats (*p* < 0.001), which was an indicator of a severe reference memory deficit (Table 3).

### 3.2. Aβ 25–35 i.c.v. Injection in Dams Disrupted Development in Offspring, an Effect Reversed by BM-MSC and GSI-953 Treatments

The hippocampus of offspring from rats that received saline injections had a normal histological structure across the testing period, as shown in Figure 3(A1–A3), Figure 4(A1–A3) and Figure 5(A1–A3). Typically, the dentate gyrus (DG) and cornu ammonis (CA) were visible. The CA comprised pyramidal cell, polymorphic, and molecular layers, which were rich in blood capillaries and glial cells. The four CA subfields (CA1, CA2, CA3, and CA4) were also present. CA1 contained 5–6 layers of small pyramidal cells with vesicular nuclei, whereas 3–4 pyramidal cell layers were found in CA2. Offspring from rats injected with saline also exhibited a typical DG composed of an upper layer of weakly stained granular cells with vesicular nuclei and a lower layer of immature granular cells situated in the subgranular zone. CA4 was found between the DG upper and lower limbs and also comprised molecular and polymorphic layers, of which the thickness increased from PND7 to PND21.

The CA1 and CA2 subfields in offspring from G2 rats exhibited a decreased thickness and a smaller disorganized pyramidal cell layer with loosely packed cells. Specifically, at PND7 (Figure 3(B1,B2)) and PND21 (Figure 5(B1,B2)), some cells appeared dark and shrunken and had pyknotic nuclei with pericellular halos (Figure 4(B1,B2)). Other regions were devoid of cells, particularly at PND14 (Figure 4(B2)). Additionally, swollen glial cells with vacuoles and dilated blood vessels were present in the molecular layer (Figure 4(B3)). The granular cell layer of the DG showed notable vacuolation, a significant delay in the process that caused separation, the enlargement of the subgranular zone, and a small number of immature cells (Figure 3(B3) and Figure 4(B3)). The DG also contained a few granule cell bodies that appear to be normal, particularly at PND14, and dark, shrunken granule cell bodies with pyknotic nuclei and pericellular halos (Figure 4(B3)). Additionally, the thickness of the granule cell, molecular, and polymorphic layers in G2 rat offspring was noticeably decreased (Figure 5(B3)). These histopathological anomalies were the most visible at PND14 than at previously investigated time points. The DG’s granular cells grew more tightly packed with age as indicated in Figure 5(B3), but the molecular and polymorphic layers seemed less cellular and more fibrous.

The treatment with BM-MSCs or GSI-953 improved the pathological alterations caused by Aβ 25–35 as the number of injured neuronal cells was reduced at all examined PNDs. Only a few of the tightly packed, uniformly distributed pyramidal cell bodies in the CA1 and CA2 were shrunken with pyknotic nuclei, whereas the majority appeared to be normal, as revealed in Figure 3(D1,D2,E1,E2), Figure 4(D1,D2,E1,E2) and Figure 5(D1,D2,E1,E2). Additionally, BM-MSCs and GSI-953 restored the granule cell layer thickness, whereas the molecular and polymorphic layer thickness remained diminished. Dilated blood capillaries and glial cells were detected in the polymorphic and molecular layers. Only a few of the granule cell bodies in the granule cell layer were black with pyknotic nuclei, whereas the majority of the DG granule cells appeared to be normal, as illustrated in Figure 3(D3,E3), Figure 4(D3,E3) and Figure 5(D3,E3). The subgranular zone exhibited a sizable number of immature neurons. Glial cells and blood capillaries were present in the polymorphic and molecular layers.

### 3.3. Amyloid Beta Deposition Confirmed by Congo Red Staining: Figure 6, Figure 7, Figure 8, Figure 9, Figure 10 and Figure 11

Congo red staining displayed amyloid beta deposition in the CA1, CA2, and DG of the hippocampus in the amyloid beta 25–35-injected dams group and also in their newborns at all tested ages, as shown in Figure 6(B1–B3), Figure 7(B1–B3), Figure 8(B1–B3), Figure 9(B1–B3), Figure 10(B1–B3) and Figure 11(B1–B3). The amyloid deposits appeared as intracellular red deposits. AD dams treated with MSCs or GSI-953 showed the decreased dimensions of amyloid deposits in all tested regions at all examined ages. In addition, the same results were observed in the hippocampus of the newborns at all tested ages, as shown in Figure 6(C1–C3,D1–D3), Figure 7(C1–C3,D1–D3), Figure 8(C1–C3,D1–D3), Figure 9(C1–C3,D1–D3), Figure 10(C1–C3,D1–D3) and Figure 11(C1–C3,D1–D3).
Figure 6Photomicrographs of Congo red-stained CA1, CA2, and DG of hippocampus tissue of dams at PND7. (**A**) Saline-injected group exhibiting a normal histology structure of hippocampus with normal neuronal cells; (**B**) AD group exhibiting marked Aβ deposition in hippocampus (arrow); (**C**) AD treated with MSCs group showing a normal appearance of the hippocampus with intact neuronal cells mild multifocal Aβ deposition (arrow); (**D**) AD treated with the GSI-953 group showing a normal histological appearance of the hippocampus with intact neuronal cells (arrow) as well as no Aβ deposition. *n* = three animals for each group.
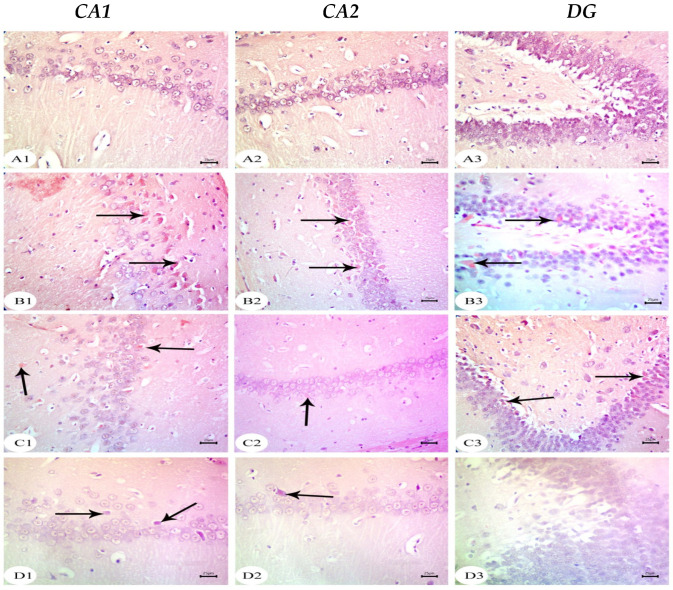

Figure 7Photomicrographs of Congo red-stained CA1, CA2, and DG of the hippocampus tissue of dams at PND14. (**A**) Saline-injected group exhibiting a normal histology structure of the hippocampus with normal neuronal cells; (**B**) AD group exhibiting marked Aβ deposition in hippocampus (arrow); (**C**) AD treated with the MSCs group showing a normal appearance of the hippocampus with intact neuronal cells with mild Aβ deposition (arrow); and (**D**) AD treated with the GSI-953 group showing the normal histological appearance of the hippocampus with intact neuronal cells (arrow) as well as no Aβ deposition. *n* = three animals for each group.
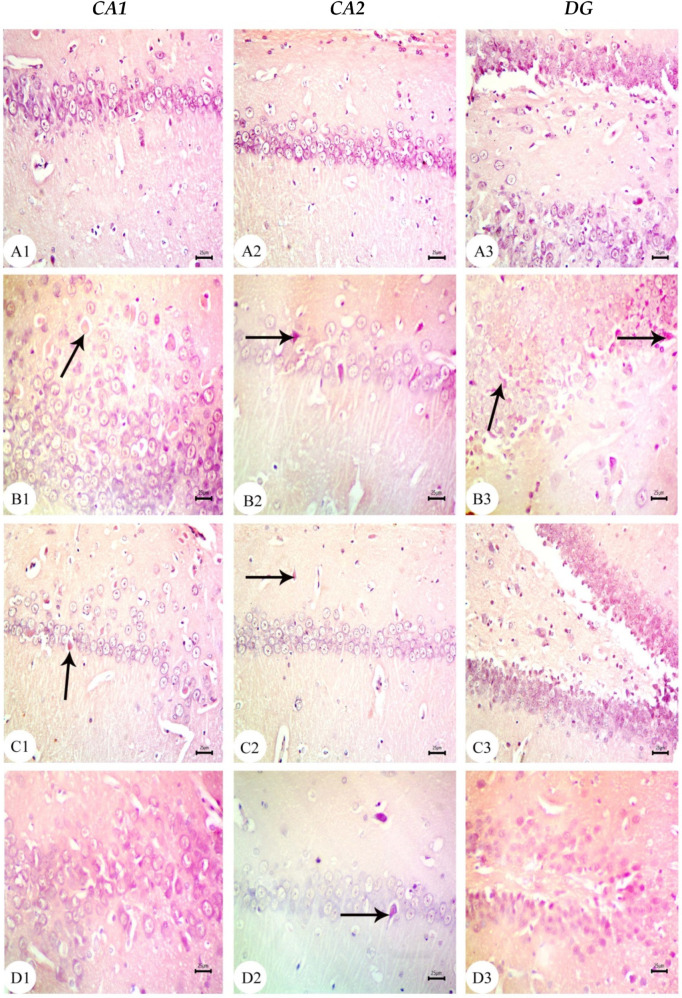

Figure 8Photomicrographs of Congo red-stained CA1, CA2, and DG of the hippocampus tissue of dams at PND21. (**A**) Saline-injected group exhibiting a normal histology structure of the hippocampus with normal neuronal cells; (**B**) AD group exhibiting marked Aβ deposition in the hippocampus (arrow); (**C**) AD treated with MSCs group showing a normal appearance of the hippocampus with intact neuronal cells with mild multifocal Aβ deposition (arrow); and (**D**) AD treated with the GSI-953 group showing a normal histological appearance of the hippocampus with intact neuronal cells as well as no Aβ deposition. *n* = three animal for each group.
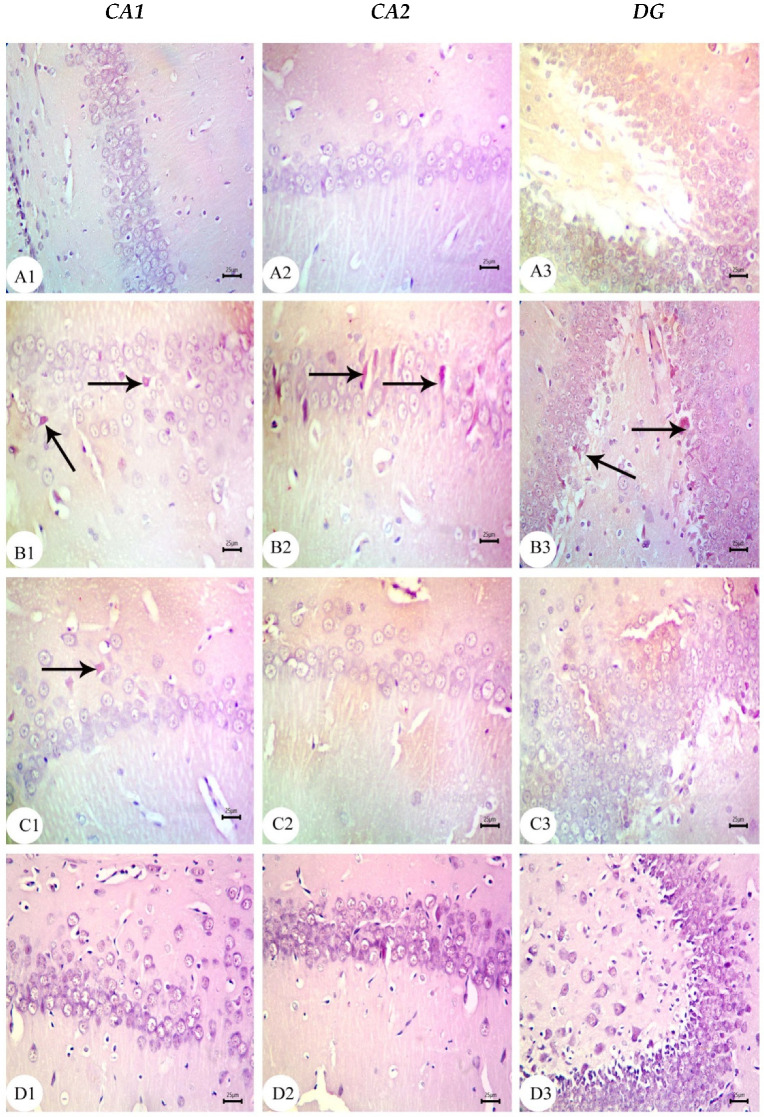

Figure 9Photomicrographs of Congo red-stained CA1, CA2, and DG of newborns’ hippocampus tissue at PND7. (**A**) Saline-injected group exhibiting the normal histology structure of the hippocampus with normal neuronal cells; (**B**) AD group exhibiting marked Aβ deposition in the hippocampus (arrow); (**C**) AD treated with MSCs group showing a normal appearance of the hippocampus with intact neuronal cells with mild Aβ deposition (arrow); (**D**) AD treated with the GSI-953 group showing normal histological appearance of hippocampus with intact neuronal cells (arrow) as well as no Aβ deposition. *n* = three animals for each group.
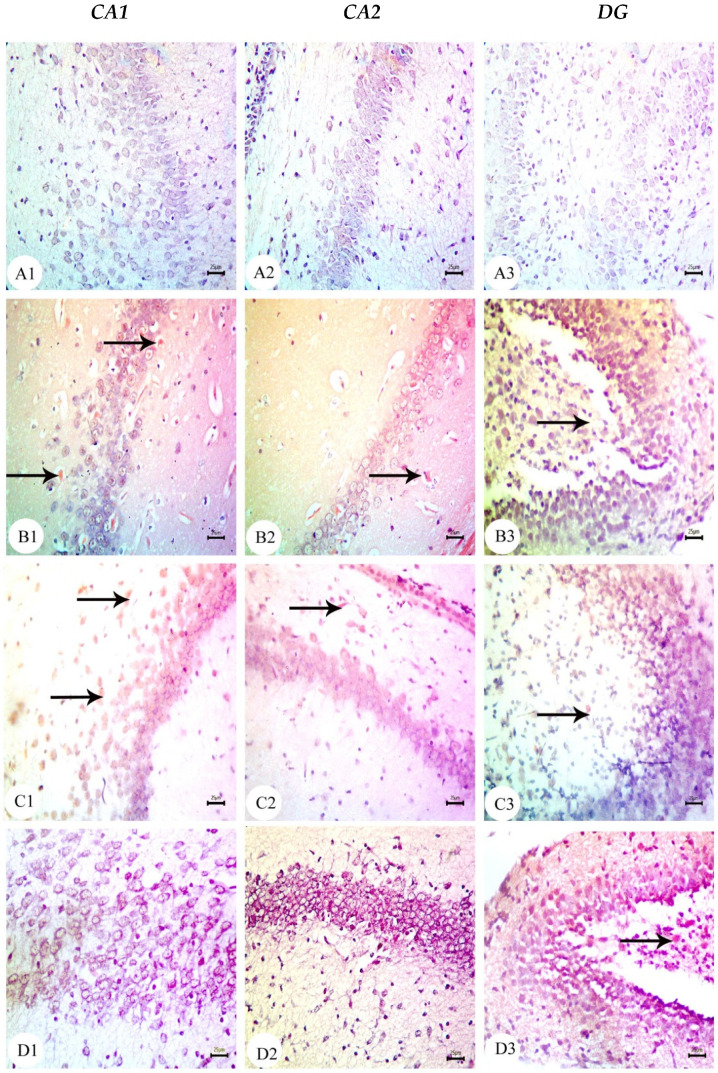

Figure 10Photomicrographs of Congo red-stained CA1, CA2, and DG of the hippocampus tissue of newborns at PND14. (**A**) Saline-injected group exhibiting a normal histology structure of the hippocampus with normal neuronal cells; (**B**) AD group exhibiting marked Aβ deposition in the hippocampus (arrow); (**C**) AD treated with MSCs group showing a normal appearance of the hippocampus with intact neuronal cells with extremely low Aβ deposition (arrow); and (**D**) AD treated with GSI-953 group showing a normal histological appearance of the hippocampus with intact neuronal cells (arrow) as well as no Aβ deposition. *n* = three animals for each group.
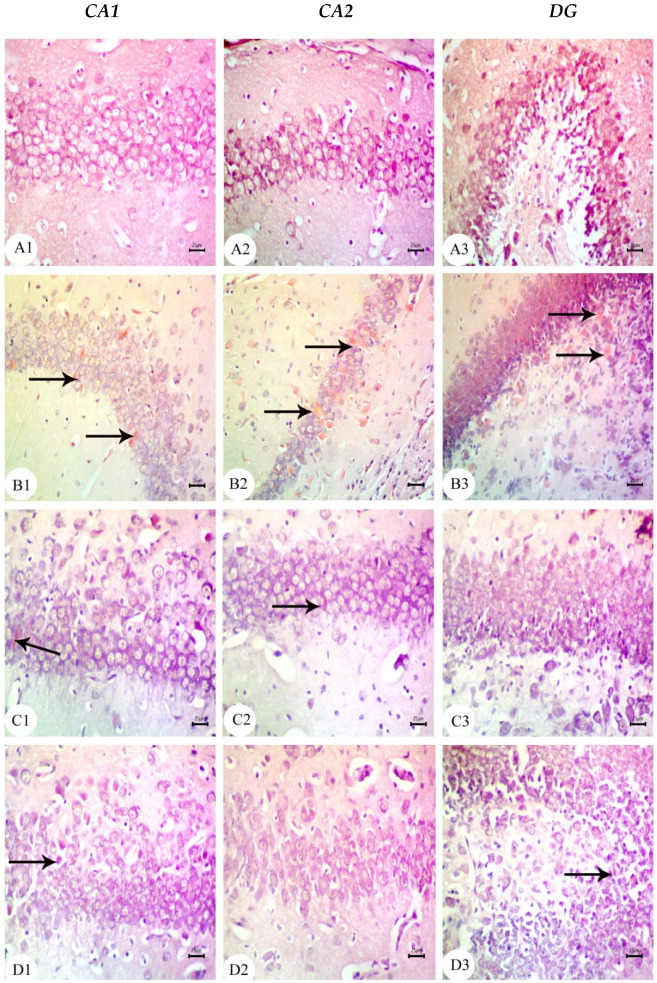

Figure 11Photomicrographs of Congo red-stained CA1, CA2, and DG of the hippocampus tissue of newborns at PND21. (**A**) Saline-injected group exhibiting a normal histology structure of the hippocampus with normal neuronal cells; (**B**) AD group exhibiting marked Aβ deposition in the hippocampus (arrow); (**C**) AD treated with the MSCs group showing a normal appearance of the hippocampus with intact neuronal cells with slight Aβ deposition (arrow); and (**D**) AD treated with the GSI-953 group showing a normal histological appearance of the hippocampus with intact neuronal cells as well as no Aβ deposition. *n* = three animals for each group.
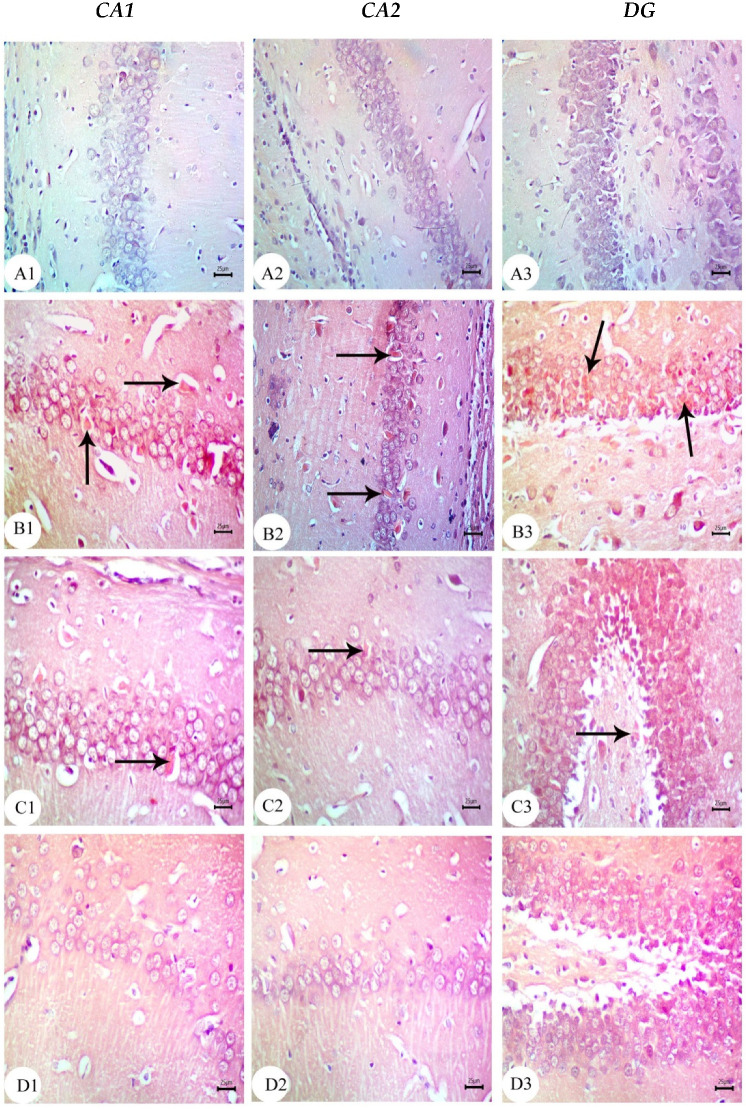


### 3.4. BM-MSC and GSI-953 Treatments Restored near Normal Microglial Cell Counts, Soma Size, and Dendrite Length in the Hippocampus of Offspring from Aβ 25–35-Injected Dams

As a result of Aβ deposition, microglial cells become overly activated, which dramatically accelerates the progression of AD. Thus, we measured the number of activated microglial cells in the newborn hippocampal area (Figure 12, Figure 13 and Figure 14) using Iba-1 immunostaining. The number of microglial cells (Figure 14a) and the process length (Figure 14c) significantly (*p* < 0.001) increased in offspring from the saline-injected group (G1) from PND7 to PND21. Between PND7 and PND21, the offspring from the Aβ 25–35-injected group (G2) had significantly more Iba-1-positive cells, larger microglial soma, and a shorter process length (all *p* < 0.001) compared with those in age-matched G1 neonates (Figure 14a–c). According to qualitative histological analyses, BM-MSC and GSI-953 treatments significantly (*p* < 0.001) increased the processes length at PND7, PND14, and PND21 and restored the microglial number and soma size. Along with these main effects, the two-way ANOVA revealed a significant (*p* < 0.001) interaction between the therapy group and neonatal age, suggesting a return to normal hippocampal development.

### 3.5. BM-MSC and GSI-953 Treatments Restored Normal Serum Levels of Proinflammatory, Prodegenerative, and Neuroprotective Factors in Offspring of Aβ 25–35-Injected Dams

#### 3.5.1. Effects on Neuroinflammatory Cytokine Serum Levels

Serum levels of interleukin-1β (IL-1β) and tumor necrosis factor-α (TNF-α) in offspring at various PNDs to assess the effectiveness of BM-MSCs and GSI-953 against Aβ 25–35-induced neuroinflammation. Serum levels of IL-1β and TNF-α were significantly (*p* < 0.001) reduced in offspring from dams treated with BM-MSC or GSI-953 at all tested PNDs. IL-1β and TNF-α concentrations were significantly (*p* < 0.001) elevated in the offspring of the AD group compared with that in the offspring from the-saline-injected group at all PNDs examined (Figure 15). Particularly, from PND7 to PND21, AD offspring had considerably higher levels of IL-1β and TNF-α (*p* < 0.001). Two-way ANOVA showed a significant group × age interaction (F8, 75 = 57.186, *p* < 0.001) and a significant main effect of the treatment on IL-1β serum level differences among groups (F4, 75 = 1212.932, *p* < 0.001) and PNDs (F2, 75 = 39.079, *p* < 0.001). Additionally, there was a significant treatment group × age interaction (F8, 75 = 273.126, *p* < 0.001) and a significant main effect of the treatment type (F4, 75 = 1743.864, *p* < 0.001) and neonatal age (F2, 75 = 91.433, *p* < 0.05) on TNF-α serum levels.

#### 3.5.2. Effects on Glycogen Synthase Kinase-3β (GSK-3β) and BDNF Serum Levels

We assessed the GSK-3β and BDNF serum levels in offspring to evaluate the neuroprotective efficacies of MSCs and GSI-953 against neurotoxicity and tau hyperphosphorylation induced by the Aβ 25–35 i.c.v. injections (Figure 16). BM-MSC and GSI treatments significantly prevented these alterations at all postnatal ages (all *p* < 0.001 vs. values in G2 offspring), although serum GSK-3β levels were significantly enhanced and BDNF concentrations were significantly decreased in the serum of G2 offspring compared with that in G1 offspring at all tested PNDs. Additionally, in G2 offspring, the GSK-3β serum levels were higher and the BDNF concentrations were lower at PND21 compared with those found at younger ages (both *p* < 0.001), indicating a progressive degenerative process. Two-way ANOVA showed a significant main effect of the treatment group (F4, 75 = 1099.851, *p* < 0.001) and an insignificant main effect of the postnatal age (F2, 75 = 0.153, *p* > 0.05) on the GSK-3β serum levels. Significant group × time interaction (F8, 75 = 119.446, *p* < 0.001), main effect of the postnatal age (F2, 75 = 246.179), and main effect of the treatment group on BDNF serum levels were observed.

### 3.6. BM-MSC and GSI-953 Treatments Restore Normal Expression Levels of Neuroprotective and Proapoptotic Genes in the Hippocampus of Offspring from Aβ 25–35-Injected Dams

#### 3.6.1. Effects on BDNF, Caspase-3, TGFβ, NF-κB, and TNFR Gene Expression in the Neonatal Hippocampus

The expression of the genes coding for BDNF, caspase-3, TGF-β, NF-κB, and TNFR in the hippocampus of newborn rats (Figure 17a–e) was assessed using qRT-PCR. The expression levels of BDNF mRNA were significantly (*p* < 0.001) lower in G2 offspring at all tested ages compared with that in G1 offspring, and this effect was dramatically (*p* < 0.001) reversed by the BM-MSC and GSI-953 treatments. The expression of TNFR, caspase-3, and NF-κB significantly increased with the age of G2 offspring. Additionally, two-way ANOVA revealed a significant main effect of the treatment group, a significant main effect of the neonatal age, and a significant treatment group × age interaction (*p* < 0.001). Moreover, the G2 newborn hippocampus had considerably (*p* < 0.001) higher levels of caspase-3, TGF-β, NF-κB, and TNFR mRNAs than the G1 neonatal hippocampus.

Caspase-3, TGF-β, NF-κB, and TNFR gene expression was significantly (*p* < 0.001) down-regulated in the offspring of Aβ 25–35-injected dams that received BM-MSCs or GSI-953 therapy compared with that in the neonates of Aβ 25–35-injected dams at all postnatal ages. The two-way ANOVA showed a significant (*p* < 0.001) main effect of the treatment group at all examined ages, and the expression levels of caspase-3, TGF-β, NF-κB, and TNFR mRNAs increased with the neonatal age.

#### 3.6.2. Effects on Phosphorylated Tau and APP Protein Levels in Neonatal Hippocampus

Tau phosphorylation is an essential pathogenic feature of AD and influences disease progression. BM-MSC and GSI-953 treatments significantly (*p* < 0.001) reduced tau phosphorylation in the hippocampus at all ages investigated compared with that in the offspring of G2 dams. However, we found a significant increase in phosphorylated tau protein in the hippocampus of offspring from G2 dams compared with that of G1 offspring (Figure 18a,b). The two-way ANOVA revealed that the treatment group’s main effect (*p* < 0.001) and the group x age interaction (*p* < 0.001) were significant. Similarly, the APP protein levels were significantly (*p* < 0.001) higher in the hippocampus of the G2 offspring compared with that in G1 offspring; however, the MSC and GSI-953 treatments significantly (*p* < 0.001) reversed this elevation (Figure 18a,b and Appendix A) at all tested ages. Additionally, a significant main effect of the treatment group and a significant group × age interaction were found using two-way ANOVA (both *p* < 0.001).

## 4. Discussion

The current study demonstrates, for the first time, that the prenatal administration of BM-MSCs and GSI-953 can prevent not only cognitive deficiency but also pathological alterations typical of AD, such as tau pathologies and neuroinflammation later in infants. These results raise the novel hypothesis that AD can be a developmental disability with a late-life phenotype, and that altering the brain’s molecular environment through the administration of the right care during a crucial stage of brain development can be a successful therapeutic approach for the later prevention of AD and other related conditions [14]. Establishing effective AD treatment is hampered by the fact that degenerative processes start years before clinical symptoms appear, at which point much of the damage caused by the disease may be irreparable. Therefore, the best method to lessen the illness burden is to produce safe yet efficient prophylactic medicines. Investigating therapies that are effective against AD toxicity in the developing and young adult brain may be a useful strategy [14].

The mouse neural tube is fully formed by gestational day 9.5 × 10^−10^, whereas the rat neural tube is formed by 10.5 × 10^−11^ [40]. We show that the BM-MSC injection or GSI-953 oral administration prevents brain maldevelopment following maternal Aβ 25–35 injection during gestation, suggesting that this rapid effect may be particularly helpful in testing preventive therapies against AD. We investigated the effects of BM-MSCs and GSI-953 on neuroinflammatory signaling, microglial activation, hippocampal cell migration, neuronal development, neurotrophic factor signaling, and apoptosis to elucidate their protective mechanisms.

In the current investigation, we elucidated the potential underlying mechanisms of action for BM-MSCs and GSI-953 as treatments for Alzheimer’s disease brought on by a single intracerebroventricular injection of the Aβ 25–35 rat model (Figure 19).

Memory impairment is the earliest clinical sign of AD [41]. One of the most common ways to induce AD-like disease in animal models is through the intracerebral injection of Aβ 25–35, which then forms fibril sheets that mimic the toxicity of Aβ1–40 [41,42]. According to previous publications [43,44], a single i.c.v. injection of Aβ 25–35 into adult female rats significantly impaired their working and long-term memories, as shown by their poor performance in the Y-maze and novel object recognition tests, respectively. The injection of Aβ 25–35 resulted in thinner DG and a pyramidal cell layer of the CA1 and CA2, dilated blood vessels, and notable histological alterations including shrunken cells with pyknotic nuclei and degenerated neurons in the offspring hippocampus. From PND7 to PND21, the extent of these pathological symptoms increased, which is consistent with a prior study showing hippocampal thinning in patients with moderate cognitive impairment, which is considered a precursor to clinical AD [45]. Maternal treatment with BM-MSC and GSI-953 corrected these histological alterations. GSI-953 may inhibit the plaque load, whereas BM-MSCs’ protective impact may be explained by a metabolic compensation for abnormal mitochondria [46,47]. This may lessen the inflammation and other detrimental processes that may occur as a result of plaque formation.

The pathophysiology of AD includes synaptic failure, oxidative stress, neuroinflammation, and mitochondrial dysfunction in addition to the build-up of amyloid plaques and NFTs [48]. Microglial cells are the resident macrophages of the central nervous system [35], and pathological insults result in their activation, which in turn causes the recruitment of external inflammatory cells and local inflammatory signaling [49]. Additionally, Aβ-induced microglial activation and subsequent inflammatory responses accelerate Aβ deposition [50], leading to self-sustained neurodegeneration. According to other research [42,43], Aβ 25–35 injection into the dam lateral ventricle greatly increased the number of activated microglial cells in the newborn’s hippocampus. This may be due to the phagocytic activity of activated microglial cells near Aβ deposits [51,52]. This response was reversed by BM-MSC injection, which also agrees with earlier findings [1,53]. Panchenko et al. [54] demonstrated that intravenously administered BM-MSCs are capable of crossing the blood–brain barrier and entering the hippocampus DG and temporal cortex. Additionally, we demonstrated, for the first time, that oral GSI-953 treatment can reduce microglial cell activation induced by the i.c.v. injection of Aβ 25–35. This effect may be caused by the inhibition of γ-secretase activity, resulting in a decrease in APP to Aβ conversion [18].

Aβ peptide is produced by the sequential cleavage of APP by β-secretase and γ-secretase [15]. APP concentration was noticeably higher in the hippocampus of offspring from Aβ 25–35-injected dams compared with that in the offspring from dams that received saline injections, but this increase was abolished by maternal prenatal treatments with BM-MSCs or GSI-953. Begacestat inhibits APP synthesis via direct action on gamma secretase inhibition and an increasing beta c-terminal fragment causing a reduction in Aβ production [18]. Begacestat, the initial version of the Notch-sparing GSIs, binds with the adenosine triphosphate terminal of the γ-secretase complex; moreover, the mode of action of second-generation Notch-sparing GSIs is unknown [55]. According to the new hypothesis, these GSIs show pharmacological effects by occupying the substrate binding sites on γ -secretase which are dissimilar among Notch and APP [55]. The generation of Aβ is inhibited by the powerful, specific γ-secretase inhibitor GSI-953, which has a thiophene-containing sulfonamide component. The chemical Begacestat was discovered to be equally capable of blocking both Notch intra-cellular domain (NICD) and Aβ42 synthesis in an improved cell-based technique for evaluating the γ-secretase enzyme activity against Notch and APP substrates [18]. It has been demonstrated that GSI-953 additionally inhibits the synthesis of Aβ and has an extra 15-fold selectivity towards preventing APP cleavage vs. Notch cleavage. Martone et al. [18] showed that GSI-953 inhibits APP cleavage and decreases Aβ production in vitro and in a transgenic mouse model, thus supporting the present findings. Moreover, a rapid dose-dependent drop in Aβ40 and Aβ42 levels in the brains of transgenic APP Tg2576 mice was induced by GSI-953 administration, and this shift was linked with alterations in both CSF and plasma Aβ levels.

Tau is a microtubule-associated protein that supports synaptic function and neuronal structure [56]. By activating GSK-3β, amyloid deposition leads to the hyperphosphorylation of tau kinases [57,58]. Earlier research reported a persistent increase in hyperphosphorylated tau concentration in the hippocampus following Aβ 25–35 injection [43,59] and an upregulation in GSK-3β expression and enhanced hyperphosphorylated tau levels in the newborn brain. Tau’s affinity for microtubules is decreased by its hyperphosphorylation, which consequently affects axonal transport and causes synaptic dysfunction [60,61]. This increase in hyperphosphorylated tau levels was reduced by maternal treatment with BM-MSCs or GSI-953, perhaps by decreasing GSK-3β activity [57,62].

BDNF supports neuronal development and survival by inhibiting the expression of proapoptotic proteins and increasing the levels of neurofunctional proteins through the cyclic adenosine monophosphate response element-binding protein transcription pathway [63,64]. BDNF is transmitted from the mother to the offspring through breast milk and crosses the placenta in vivo [14,65]. Maternal Aβ 25–35 injection dramatically reduced BDNF gene expression in the offspring hippocampus and decreased the serum levels of BDNF protein compared with those in the offspring of mothers who received saline injections. Similarly, pro-BDNF, mature BDNF, and the corresponding mRNAs are decreased by Aβ 25–35 injection in the parietal cortex and hippocampus [14]. On the other hand, maternal treatment with BM-MSCs and GSI-953 restored BDNF levels. This effect may result from the compensatory release of BDNF from transplanted BM-MSCs [66] or the reduction in GSK-3β activity caused by GSI-953.

By raising the levels of TNF-α, IL-1β, and NF-κB in the hippocampus, amyloid deposition drives the inflammatory response [67], which in turn stimulates APP expression and γ-secretase activity, leading to an increase in Aβ levels [68]. In response to an injection of Aβ 25–35, activated microglial cells release the proinflammatory cytokines TNF-α and IL-1β. Inactive NF-κB is kept in the cytoplasm through its binding to inhibitory protein κβ (Iκβ). Iκβ is phosphorylated and degraded in response to many types of stimulation, including those induced by stress, enabling NF-κBp65 to enter the nucleus and stimulate the release of proinflammatory cytokines [69]. The injection of Aβ 25–35 increases the elevation of proinflammatory signals, which may have been attenuated by the maternal administration of BM-MSCs or GSI-953 [57]. This idea is supported by the observations that MSC transplantation into AD mouse models reduces the release of proinflammatory cytokines such as TNF-α, IL-1β, and IL-6 from the activated microglia and astrocytes [14,54] and changes the polarization of microglia from the proinflammatory M1 phenotype to the immunomodulatory M2 phenotype [27]. Additionally, MSCs can directly secrete fibrogenic factors (TGF-β), neurotrophic factors (BDNF, insulin-like growth factor-1 (IGF-1), nerve growth factor (NGF)) and anti-inflammatory cytokines (IL-4, IL-10 [66]. TGF-β engages in a number of pathways that control amyloid metabolism, immunoregulation, and neuroprotection, whereas granulocyte-macrophage colony-stimulating factor (GM-CSF) attracts peripheral monocytes that are then stimulated by Aβ deposits and accelerates the clearance of Aβ [70]. According to the present data, the ability of GSI-953 to control inflammatory responses induced by Aβ 25–35 injection cannot be explained by anything other than a direct decrease in Aβ deposition.

In line with Zhang et al. [71], who reported that Aβ 25–35 stimulates proapoptotic Bax (Bcl-2-associated X protein) and promotes the inhibition of anti-apoptotic Bcl2 (B-cell lymphoma 2), resulting in elevated caspase-3 activity and neural death, the treatment of pregnant rats with BM-MSCs or GSI-953 significantly suppressed the Aβ 25–35-induced upregulation of the apoptosis effector cleaved caspase-3 in neonatal hippocampus. Additionally, cleaved caspase-3 activity is inhibited by transplanted MSCs [72] secreting seldin and survivin [73], proteins that bind caspase-3 and stop the subsequent death cascade [74].

## 5. Conclusions

The cognitive deficits induced by the i.c.v. injections of Aβ 25–35 was improved by treating adult female rats with i.v. injections of BM-MSCs or the oral administration of the γ-secretase inhibitor GSI-953 (Begacestat). Additionally, prenatal BM-MSC and GSI-953 treatments protected against hippocampus maldevelopment in neonates. These protective effects in the neonatal rat hippocampus were linked to a reduced expression of APP and hyperphosphorylated tau protein, the suppression of microglial activation, the decreased expression of proinflammatory cytokines (IL-1β and TNF-α), and the down-regulation of brain levels of caspase-3, NF-κB, and TGF-β (Figure 13). Moreover, the down-regulation of BDNF expression and increased GSK-3β activity induced by Aβ 25–35 were reversed by BM-MSC and GSI-953 treatments. Therefore, these therapies seem to prevent Aβ 25–35-induced neurodegeneration and cognitive impairments by preventing the formation of Aβ plaque and NFTs as well as the resulting neuroinflammation. However, additional investigations are needed to determine the precise mechanism responsible for the anti-inflammatory effect of GSI-953 and BM-MSCs.

## Figures and Tables

**Figure 3 biology-12-00905-f003:**
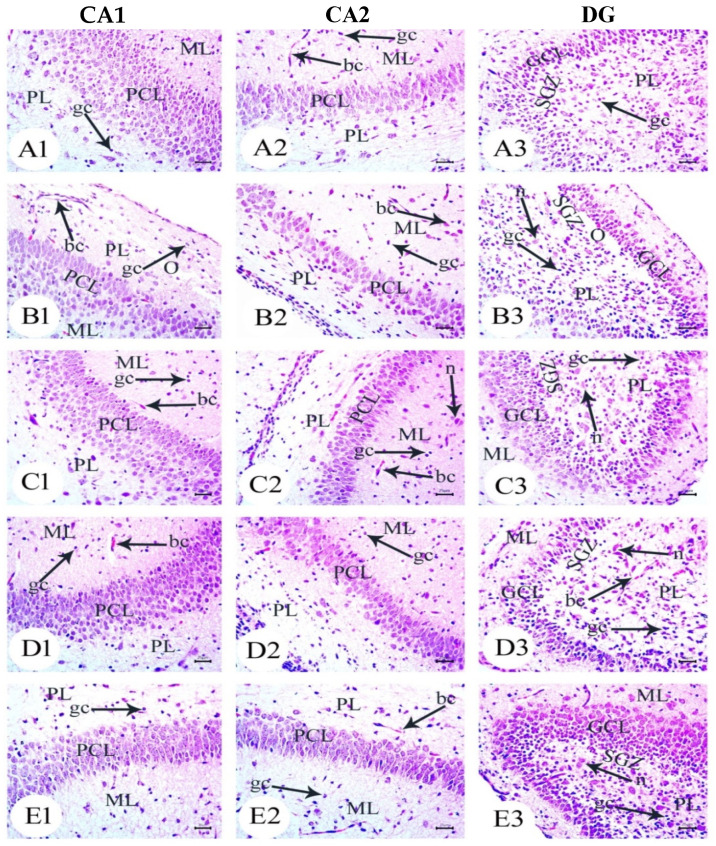
Photomicrographs of hematoxylin and eosin (H&E) stained sections of neonatal hippocampus regions of CA1, CA2, and DG at PND7 of the following groups: (**A**) saline-injected group which illustrated the normal organization of cells of the hippocampus with a normal histological structure of CA1 and CA2 of corniu ammonis that consisted of pyramidal cell layers (PCLs) containing normal pyramidal cells (PCs) with a vesicular nuclei and molecular layer (ML) and a polymorphic layer (PL), both of which were rich with blood capillaries (bc) with glial cells (gc). Dentate gryus (DG) showed normal structure of granular cell layer (GCL) that contained closely packed granular cells with a normal subgranular zone (SGZ); in addition, the ML and PL of DG contained normal glial cell (gc) with blood capillaries (bc). (**B**) The Aβ 25–35-injected group demonstrated a decreasing thickness in the PCLs of CA1 and CA2 and reducing in the thickness and number of cells forming GCL, in addition to the distortion of SGZ with a marked separation and odema (O). Furthermore, the ML and PL of both CA and DG showed an enlarged glial cell (gc) with dilated blood capillaries (bc). (**C**) The DMEM group showed a normal histological structure of the CA1, CA2, and DG regions of hippocampus. (**D**,**E**) AD treated with MSCs and AD treated with GSI-953, respectively, showed a normal structure of CA as illustrated by normal PCL containing pyramidal cells with vesicular nuclei, restoring the normal thickness of CA and DG, ML and PL showed normal glial cell (gc) with normal blood capillaries (bc). (scale bar = 25 µm). *n* = four animals for each group.

**Figure 4 biology-12-00905-f004:**
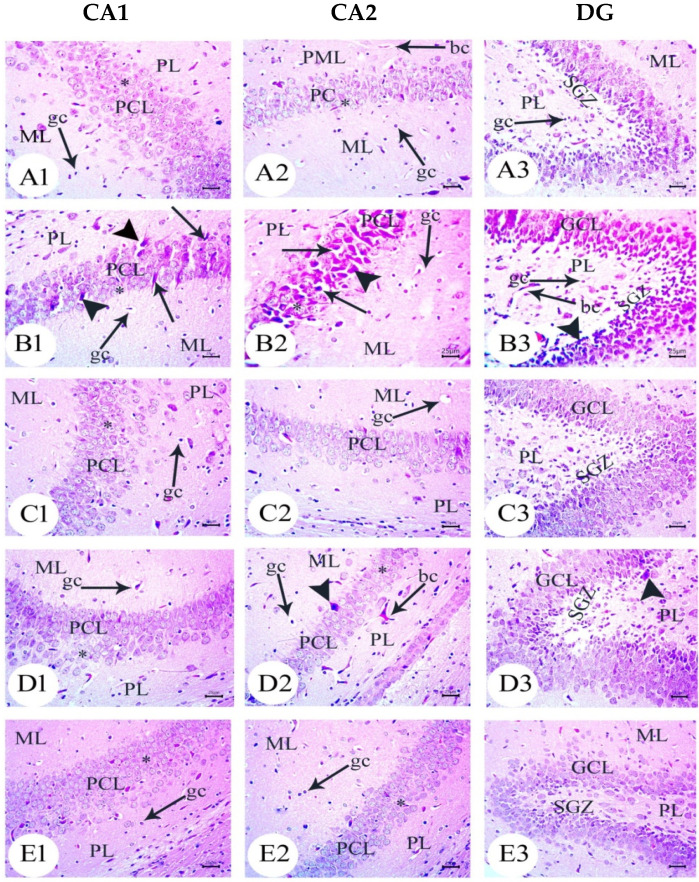
Photomicrographs of hematoxylin and eosin (H&E)-stained sections of neonatal hippocampus regions of CA1, CA2, and DG at PND14 of the following groups: (**A**) the saline-injected group which illustrated the normal organization of cells of the hippocampus with a normal histological structure of CA1 and CA2 of corniu ammonis that consisted of pyramidal cell layers (PCLs) containing normal pyramidal cells (PCs) with a vesicular nuclei and molecular layer (ML) and a polymorphic layer (PL), both of which were rich with blood capillaries (bc) with glial cells (gc). Dentate gryus (DG) showed a normal structure of granular cell layer (GCL) that contained closely packed granular cells with the normal subgranular zone (SGZ), in addition the ML and PL of DG, which contained normal glial cells (gc) with blood capillaries (bc). (**B**) In the Aβ 25–35-injected group, the CA1 and CA2 region showed that most of the cell bodies of the pyramidal neurons in PCL were disarranged and loosely packed; they appeared dark with pyknotic nuclei (arrow) and shrunken degenerated neurons with pericellular haloes (arrow head). ML and PL showed enlarged glial cells (gc) with pericellular haloes and dilated blood capillaries (bc). DG exhibited dark shrunken granule cell bodies having pyknotic nuclei (arrow head) with pericellular haloes (h). In the widening of SGZ, few immature neurons can be seen. (**C**) DMEM group showed normal histological structure of CA1, CA2, and DG regions of hippocampus. (**D**,**E**) AD treated with MSCs and AD treated with GSI-953, respectively; CA1 and CA2 region showed pyramidal cell bodies in PCL that appeared somewhat regularly arranged; a few of them were shrunken with pyknotic nuclei (arrow head) and the others were apparently normal (*). PL and ML displayed dilated blood capillaries (bc) and enlarged glial cells. DG displayed some apparently normal granule cell bodies (*), some of which appeared shrunken with pyknotic nuclei (arrowhead), and SGZ contained a considerable number of immature cells. ML and PL contained enlarged glial cells (gc) and dilated bc blood capillaries (bc). (scale bar = 25 µm). *n* = four animals for each group.

**Figure 5 biology-12-00905-f005:**
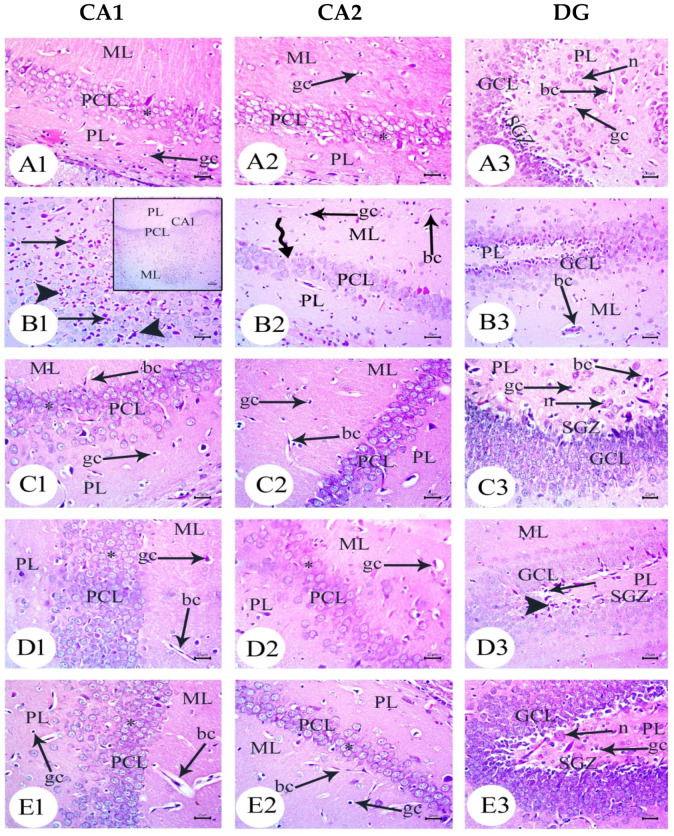
Photomicrographs of hematoxylin and eosin (H&E)-stained sections of neonatal hippocampus regions of CA1, CA2, and DG at PND21 of the following groups: (**A**) saline-injected group which illustrated the normal organization of the cells of hippocampus with a normal histological structure of the CA1 and CA2 of corniu ammonis that consisted of pyramidal cell layers (PCLs) containing normal pyramidal cells (PCs) with a vesicular nuclei and molecular layer (ML) and a polymorphic layer (PL), both of which are rich with blood capillaries (bc) with glial cells (gc). Dentate gryus (DG) showed a normal structure of the granular cell layer (GCL) that contained closely packed granular cells with a normal subgranular zone (SGZ), in addition the ML and PL of DG, which contained normal glial cells (gc) with blood capillaries (bc). (**B**) In the Aβ 25–35-injected group, the CA1 region exhibited an enlargement of PCL (square region) magnification of the square region exhibited that most of the cell bodies of the pyramidal neurons in PCL were disarranged, and appeared dark with pyknotic nuclei (arrow), shrunken degenerated neurons with pericellular haloes (arrow head). ML and PL showed enlarged glial cells (gc) with pericellular haloes and dilated blood capillaries (bc) and CA2 displayed a decreasing thickness and a number of pyramidal cells (pc) forming PCL, whilst some regions were devoid of cells (zigzag arrow). DG displayed dark shrunken granule cell bodies containing pyknotic nuclei (arrow head)) with pericellular haloes. The widening of SGZ with a few immature neurons can be seen. (**C**) The DMEM group showed a normal histological structure of CA1, CA2, and DG regions of hippocampus. (**D**,**E**) AD treated with MSCs and AD treated with GSI-953, respectively, and CA1 and CA2 region illustrated pyramidal cell bodies in PCL that appeared somewhat regularly arranged; a few of them were shrunken with pyknotic nuclei (arrow head) and the others were apparently normal (*). POL and ML displayed dilated blood capillaries (bc) and enlarged glial cells. DG displayed some apparently normal granule cell bodies, some of which appeared shrunken with pyknotic nuclei (arrowhead), and SGZ contained a considerable number of immature cells. ML and PL contained enlarged glial cells (gc) and dilated blood capillaries (bc). (scale bar = 25 µm). *n* = four animals for each group.

**Figure 12 biology-12-00905-f012:**
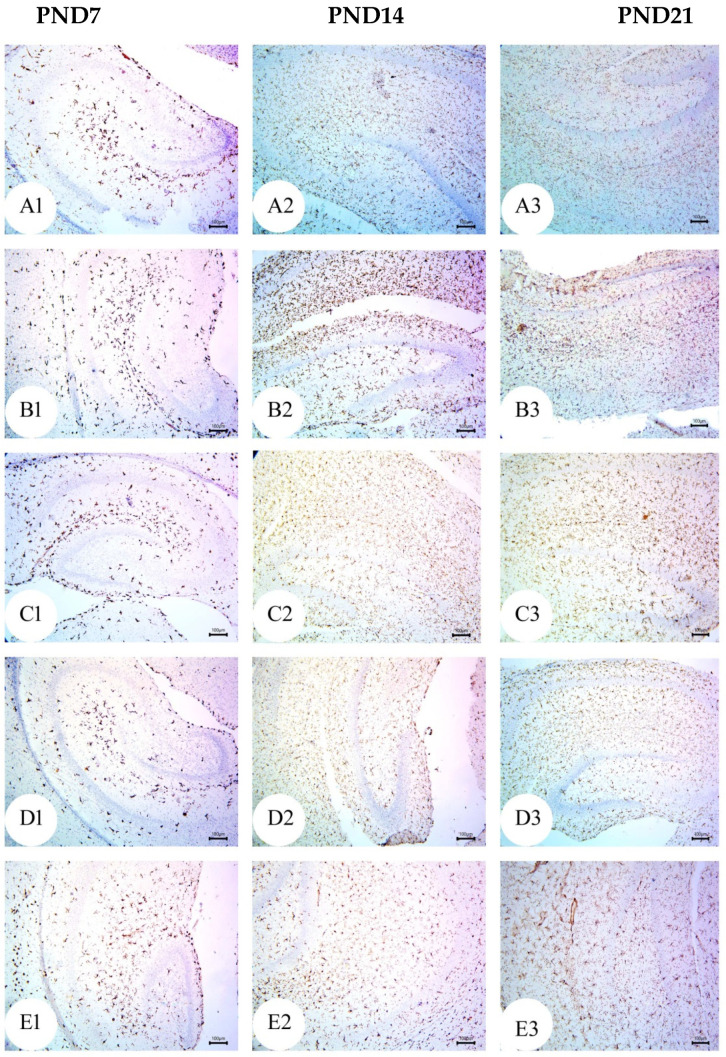
Photomicrographs of an immunohistochemistry study for neonatal hippocampus using Iba-1 to illustrate microglial cell reactivity at different postnatal days: (**A**) Saline-injected group; (**B**) Aβ 25–35 group; (**C**) DMEM group; (**D**) AD + MSCs group; and (**E**) AD + GSI. Scale bar, 100 µm. *n* = three animals for each group.

**Figure 13 biology-12-00905-f013:**
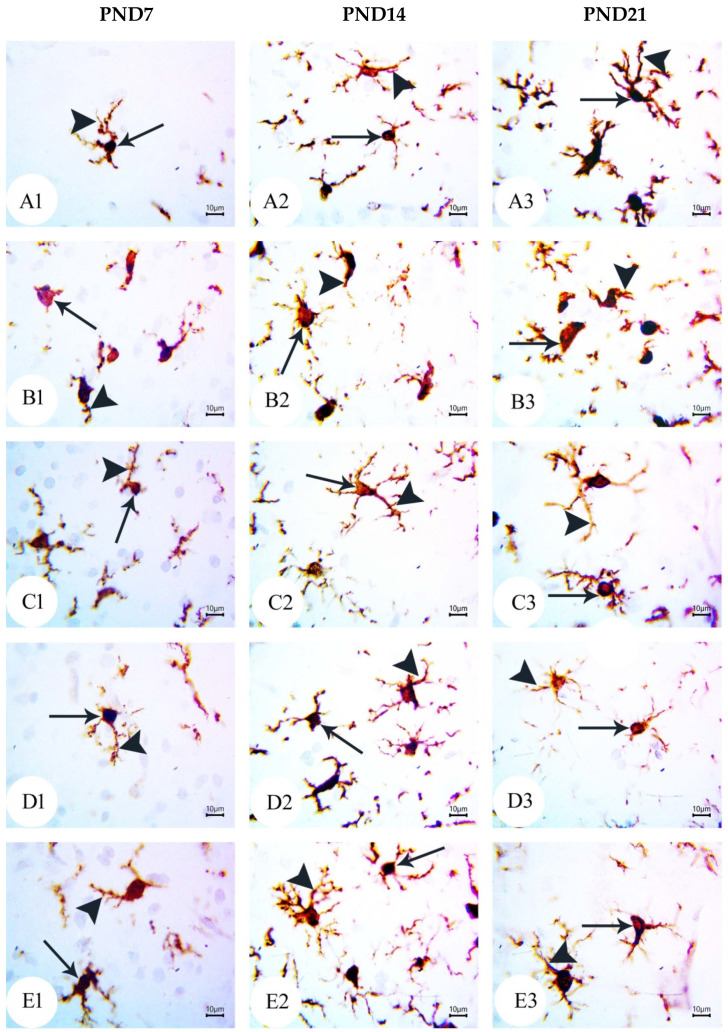
Photomicrographs of an immunohistochemistry study for the neonatal hippocampus using Iba-1 to illustrate the microglial cell soma size and dendritic length at different postnatal days: (**A**) saline-injected group; (**B**) Aβ 25–35 group; (**C**) DMEM group; (**D**) AD + MSCs group; and (**E**) AD + GSI. Scale bar, 10 µm. Arrow refers to the cell soma size, whilst the arrowhead refers to the length, thickness, and number of processes. *n*= three animals for each group.

**Figure 14 biology-12-00905-f014:**
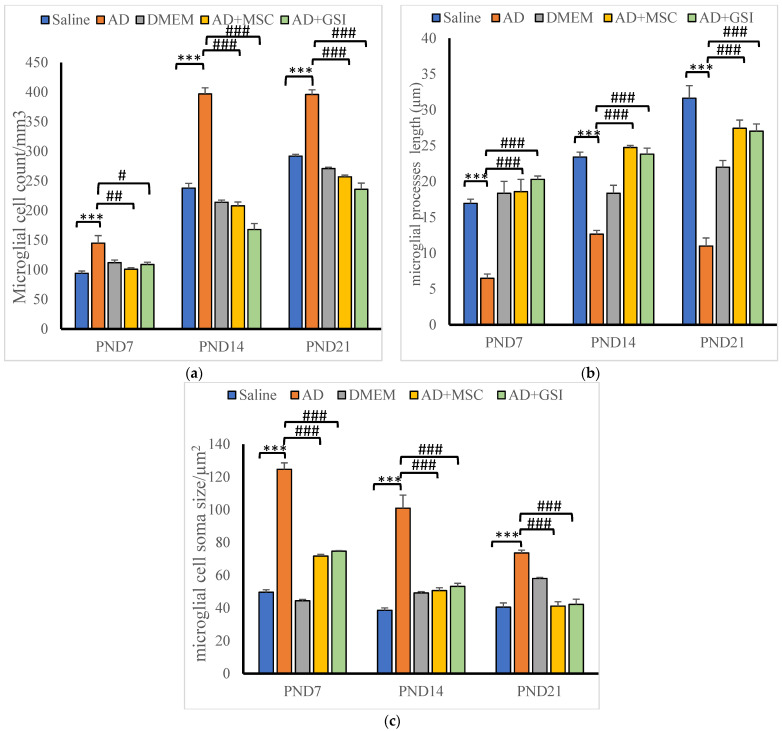
Effect of treatment with MSCs and GSI-953 on (**a**) microglial cell count/mm^3^, (**b**) dendrites length (µm), and (**c**) cell soma size (µm^2^), in newborns’ hippocampus of Aβ 25–35-induced Alzheimer’s disease in dams. Data were analyzed by two-way ANOVA followed by Turkey’s multiple comparison test. Data are expressed as mean ± SEM. *** *p* < 0.001 vs. saline-injected group. # *p* < 0.05; ## *p* < 0.01; ### *p* <0.001 vs. AD group. *n* = three animals for each group.

**Figure 15 biology-12-00905-f015:**
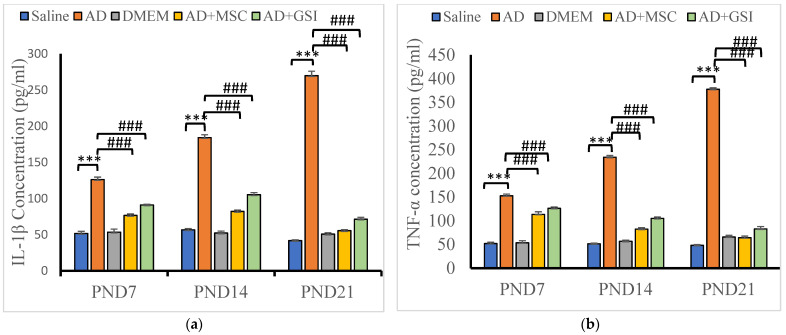
Effect of treatment with MSCs and GSI-953 on the concentration of neuroinflammatory cytokines (pg/mL): (**a**) IL-1β; and (**b**) TNF-α in newborns serum at different postnatal periods of Aβ 25–35-induced Alzheimer’s disease in dams. Data were analyzed by two-way ANOVA followed by Tukey’s multiple comparison test. Data are expressed as mean ± SEM, *** *p* < 0.001 vs. saline-injected group; ### *p* < 0.001 vs. AD group. *n*= six animals for each group.

**Figure 16 biology-12-00905-f016:**
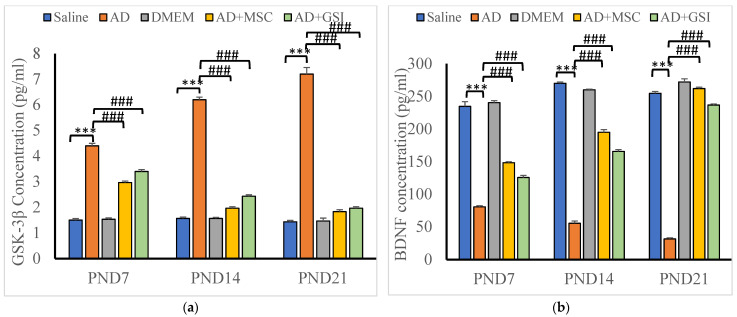
Effect of treatment with MSCs and GSI-953 on: (**a**) GSK-3β; and (**b**) BDNF concentration in newborns sera at PND7–PND21 of Aβ 25–35-induced Alzheimer’s disease in dams. Data were analyzed by two-way ANOVA followed by Tukey’s multiple comparison test. Data are expressed as mean ± SEM. *** *p* < 0.001 vs. saline-injected group. ### *p* < 0.001 vs. AD group. *n* = six animals for each group.

**Figure 17 biology-12-00905-f017:**
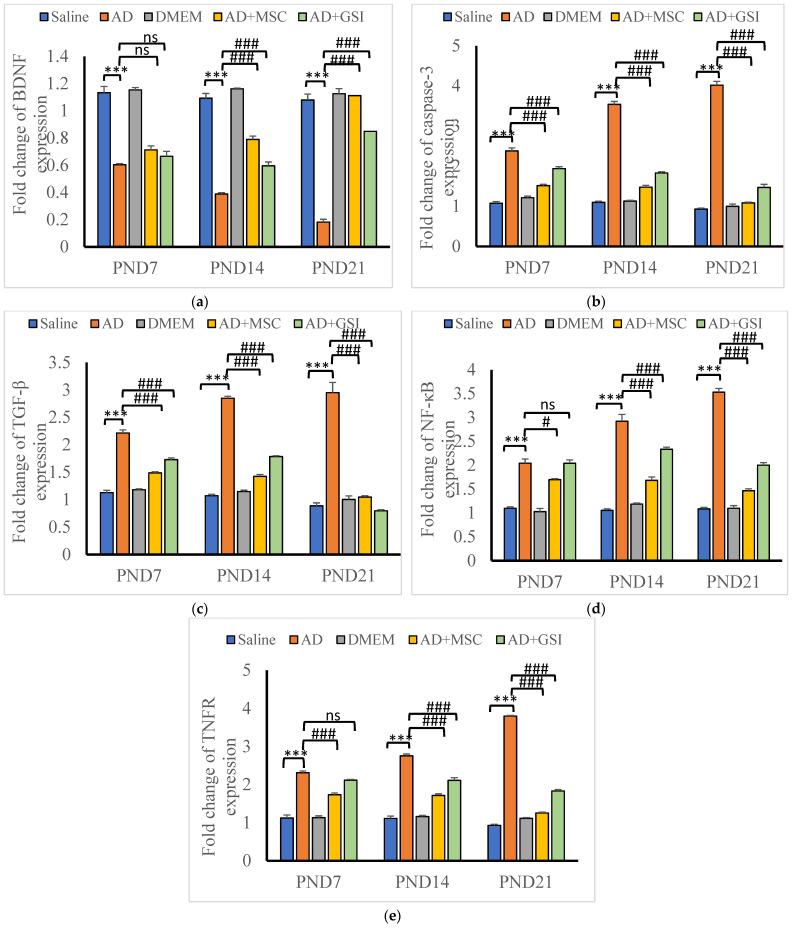
Effect of treatment with MSCs and GSI-953 on the gene expression of: (**a**) BDNF; (**b**) cleaved caspase-3; (**c**) TGFβ; (**d**) NFκB; and (**e**) TNFR in newborns’ hippocampus at different postnatal days of Aβ 25–35-induced Alzheimer’s disease in dams evaluated by qRT-PCR. Data were analyzed by two-way ANOVA followed by Tukey’s multiple comparison test. Data are expressed as mean ± SEM. *** *p* < 0.001 vs. saline-injected group. # *p* < 0.05; ### *p* < 0.001 vs. AD group. ns *p* >0.05 not significant. *n* = three animals for each group.

**Figure 18 biology-12-00905-f018:**
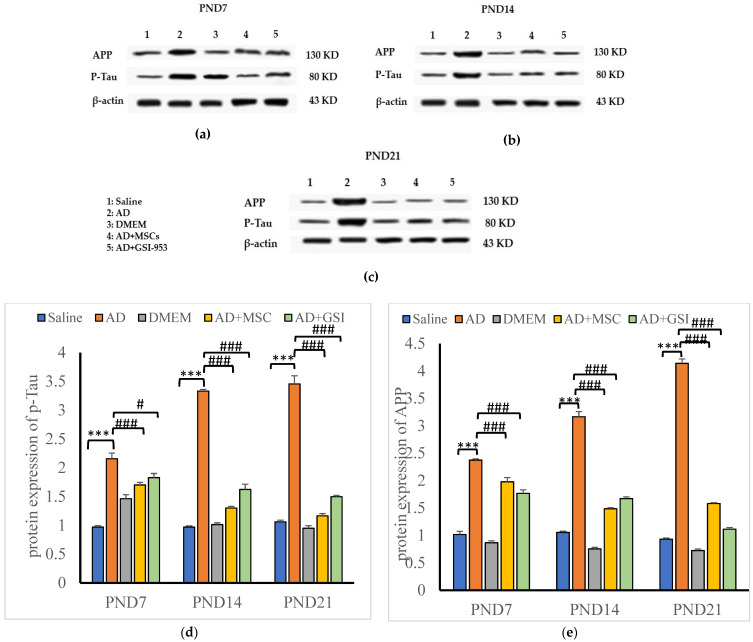
Western blots bands for APP and P-tau of newborns hippocampus at (**a**) PND7, (**b**) PND14 and (**c**) PND21. In addition, effect of treatment with MSCs and GSI-953 on: (**d**) *p*-Tau; and (**e**) APP protein relative expression levels in newborns’ hippocampus of Aβ 25–35-induced Alzheimer’s disease in dams at different postnatal days using Western blotting technique. Data are expressed as mean ± SEM. * *p* < 0.05; ***p* < 0.01; *** *p* < 0.001 vs. saline-injected group. # *p* < 0.05; ### *p* < 0.001 vs. AD group.

**Figure 19 biology-12-00905-f019:**
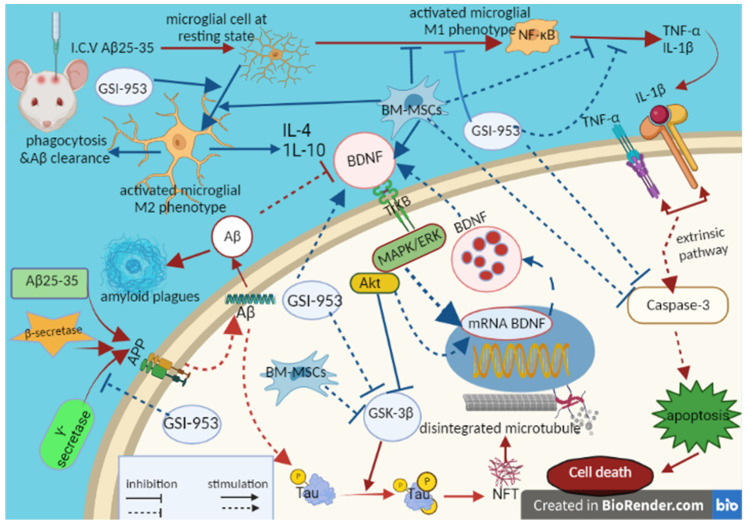
Schematic figure showing pathways of MSCs and GSI-953 against the Aβ 25–35-induced Alzheimer’s disease in neonates. Figure created by us using BioRender.com, (accessed on 1 December 2022).

**Table 1 biology-12-00905-t001:** Number of animals used in the experiment in different groups.

Sex	Purpose of Use	Number of Animals
Female	Saline-injected group	10
	AD group	10
	AD + MSCs	10
	AD + GSI-953	10
	DMEM group	10
Male	Mating with female at arrange of two female with one male	30 adult male rats

**Table 2 biology-12-00905-t002:** Sequences of primers for tested genes.

	Forward Sequence	Reverse Sequence
Caspase-3	5′-TGGTTCATCCAGTCGCTTTGT-3′	5′-CAAATTCTGTTGCCACCTTTCG-3′
TNFR	5′-GGGATTCAGCTCCTGTCAAA-3′	5′-ATGAACTCCTTCCAGCGTGT-3′
TGF-β	5′-GTCACTGGAGTTGTACGGCA-3′	5′-GGGCTGATCCCGTTGATTTC-3′
BDNF	5′-CCGGTATCCAAAGGCCAACT-3′	5′-CTGCAGCCTTCCTTGGTGTA-3′
NF-κB	5′-TTCCCT GAA GTG GAG CTA GGA-3′	5′-CATGTC GAG GAA GAC ACT GGA-3′
β-actin	5′-AGGCCC CTC TGA ACC CTA AG-3′	5′-GGA GCG CGT AAC CCT CATAG-3′

**Table 3 biology-12-00905-t003:** Effect of Aβ 25–35 injection into adult female rats before pregnancy on the behavioral test of dams, Y-maze test, and novel object recognition test. Data are presented as mean ± SEM. Behavioral tests results were analyzed by one-sample *t*-test * = *p* < 0.001, ns *p* > 0.05 insignificant effect. *n* = 6 animal per each group.

Group	Novel Object Recognition Test	Y-Maze Test	
	(Discrimination Index)	Time Percent for Novel Arm	Percent of Novel Arm Entries
Saline	0.38450 ± 0.139316	80.4667 ± 1.02783	44.5833 ± 2.451
AD	−0.53200 ± 0.093088 *	41.8667 ± 6.95608 *	44.0500 ± 4.312 ^ns^
F-value	29.919	30.135	0.012

## Data Availability

This article includes all the data generated or analyzed during this investigation.

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
