# Peer review of "Mesenchymal Stem Cells and Begacestat Mitigate Amyloid-β 25–35-Induced Cognitive Decline in Rat Dams and Hippocampal Deteriorations in Offspring"

_biology, 2023, doi:10.3390/biology12070905_

Round 1
Reviewer 1 Report (Previous Reviewer 1)
I want to express my thanks to the author who has done substantial revision upon reviewers’ comments to the manuscript. The revised version has significantly improved the quality of the manuscript. However, there are still several things should be revised before it can be accepted for publication:
1. Aβ25-35 model is not related to full length Aβ production, please provide more detail on why gamma secretase inhibitor can have such benefit to this model.
2. Since this model used oligomer Aβ25–35, please provide the western blot or electronic microscopy result to show the formation of oligomer that is used in this study.
3. In the method, the author described that G2 (n=30) was divided into G4 ad G5 (n=10), so there is 10 more left in this group as G2. This should be clearly described in the method section.
4. The author used 90 mice in the method description, and G1=10, G2=30, G3=10 so the total N=50. Please provide a table in the method to explain how the 90 mice were used in this study, and how many dams were used for each study.
5. Figure2 C& D: MSCs before injection at day 10 day of isolation. Please explain why the morphology is quite different each other, and why needs to show both C and D
6. Line 211-212 Section 2.6. Histological Analysis of the Newborn hippocampus: The whole hippocampus of neonates from each treatment group (four per group) was removed, frozen in 4% paraformaldehyde for 48 hours at 4°C. There are some issues in this description, because 4°C will not freeze, please correct it
7. Line 240-244 2.10. qRT-PCR: Total RNA was extracted from hippocampus lysates using the Direct-zol RNA Miniprep Plus kit (cat. # R2072, Zymo Research, Irvine, CA, USA) according to the man- ufacturer's instructions and reverse transcribed using the Superscript IV One-Step RT-PCR kit (cat. # 12594100, Thermo Fisher Scientific, Waltham, MA, USA) This is contradictory to previous descriptions. The author has stated that the whole hippocampus has been used for immunostaining, so please provide more information how many mice used for qRT-PCR
8. Line 373 n= four animals for each group is a duplicate of previous sentence, so please delete it
9. Line 446 n=three for each group should be “n=three animals for each group”
Acceptable
Author Response
Author response to reviewer 1 comments
Reveiwer 1 comments
I want to express my thanks to the author who has done substantial revision upon reviewers’ comments to the manuscript. The revised version has significantly improved the quality of the manuscript. However, there are still several things should be revised before it can be accepted for publication:
- Aβ25-35 model is not related to full length Aβ production, please provide more detail on why gamma secretase inhibitor can have such benefit to this model.
Author response: Thank you for your comment. As indicated in many previous publications, the β-amyloid peptide 25–35 (Aβ25–35) has the critical neurotoxic properties of the full-length Aβ1–42 (https://doi.org/10.1016/j.neures.2008.11.006). Acute intracerebroventricular (icv) injection of Aβ25–35 is involved in memory impairment (https://doi.org/10.1016/S0361-9230(03)00118-7; https://doi.org/10.1016/j.bbr.2006.09.006; https://doi.org/10.1016/j.nbd.2006.02.008). Aβ25–35 increases oxidative stress, causes neuronal damage, and decreases spatial memory in rats. It causes a significant increase of reactive astrocytosis, which was accompanied by neuronal damage in the temporal cortex and hippocampus of rats injected with Aβ25–35 (https://doi.org/10.1016/j.neures.2008.11.006). Text was added to show why gamma secretase inhibitor can have such benefit to this model in page 30 paragraph 2.
- Since this model used oligomer Aβ25–35, please provide the western blot or electronic microscopy result to show the formation of oligomer that is used in this study.
Author response: Thanks for your comment. We follow the induction protocol of previous publications (Kim et al., 2016; Zussy et al., 2013). These previous studies proved that aggregated amyloid beta after incubation is very toxic and induced Alzheimer’s disease. Amyloid precursor protein (APP), the precursor of toxic Aβ peptides, was determined by Western blot. It plays a central role in the pathophysiology of Alzheimer's disease in large part due to the sequential proteolytic cleavages that result in the generation of β-amyloid peptides (Aβ) (https://doi.org/10.1186/1750-1326-6-27). The p-Tau was also measured by Western blot. Accumulation of phosphorylated tau is a key pathological feature of Alzheimer's disease (https://doi.org/10.1093/brain/awaa223). Aβ deposition and distribution in the hippocampus was detected by Congo red staining as requested in the previous round of reviewing process.
- In the method, the author described that G2 (n=30) was divided into G4 and G5 (n=10), so there is 10 more left in this group as G2. This should be clearly described in the method section.
Author response: The 30 adult female rats were classified as 10 animals acted as AD group (negative control group), the other 10 animals acted as AD treated with MSCs while the last 10 animals acted as AD treated with GSI-953.
- The author used 90 mice in the method description, and G1=10, G2=30, G3=10 so the total N=50. Please provide a table in the method to explain how the 90 mice were used in this study, and how many dams were used for each study.
Author response: Thanks for your comment. Sorry for error, the total number of adult rats is 80 not 90. Eighty represents 50 females and 30 mature males for mating. Their classification was shown in the added table (table 1) in page 4. We used rats (not mice) in our experiment.
- Figure 2 C & D: MSCs before injection at day 10 day of isolation. Please explain why the morphology is quite different each other, and why needs to show both C and D
Author response: Thanks for your comment. C and D are two photos of MSCs at 10 days before and after washing with DMEM before injection.
- Line 211-212 Section 2.6. Histological Analysis of the Newborn hippocampus: The whole hippocampus of neonates from each treatment group (four per group) was removed, frozen in 4% paraformaldehyde for 48 hours at 4°C. There are some issues in this description, because 4°C will not freeze, please correct it
Author response: Thanks for your comment. The word “frozen” was replaced by “kept”. The correction was marked in red colour in page 6.
- Line 240-244 2.10. qRT-PCR: Total RNA was extracted from hippocampus lysates using the Direct-zol RNA Miniprep Plus kit (cat. # R2072, Zymo Research, Irvine, CA, USA) according to the manufacturer's instructions and reverse transcribed using the Superscript IV One-Step RT-PCR kit (cat. # 12594100, Thermo Fisher Scientific, Waltham, MA, USA) This is contradictory to previous descriptions. The author has stated that the whole hippocampus has been used for immunostaining, so please provide more information how many mice used for qRT-PCR
Author response: Thank you for your comment. Three samples of whole hippocampus of three newborns only from each group were used for qRT-PCR. This was declared in footnote in the title of figure 17 (Page 27). Three other samples of hippocampus from each group were used for Western blot analysis. The hippocampus of other 4 newborns form each group were fixed for histological, special (Congo red) and immunohistochemical staining.
- Line 373 n= four animals for each group is a duplicate of previous sentence, so please delete it
Author response: Thank you for your comment. The duplicated phrase was removed.
- Line 446 n=three for each group should be “n=three animals for each group”
Author response: Thank you for your comment. The phrase was changed to be n=three animals for each group”

Reviewer 2 Report (Previous Reviewer 2)
The authors have addressed most of the comments raised by this reviewer. I have few additional comments:
Comments to authors:
11. Did the authors perform Y-maze test post treatment with BM-MSCs and GSI-953? If so, the results showing any improvement in the behavior of the animals post-treatment must be provided.
22. Old comment: Most importantly, there is missing evidence of the characterization of the BM-MSCs, used in the study, considering they show “fibroblast like morphology”.
Author response: Thank you for your valuable comment. The evidence for the characterization of the BM-MSCs was included in our previous publications (https://www.mdpi.com/1424-8247/16/1/34)
New comment: Although the authors have characterized the BM-MSCs in their earlier studies, the characteristics of the cells may differ from batch to batch and the isolation conditions; therefore, the authors must have performed some analysis on the activity or immunohistochemical analysis of the cells used in this particular study which is missing. In addition, the above-mentioned article must be cited in the method section.
13. The Western blot in Figure 18a. The phospho-tau bands do not correspond with the original blot provided in the supplementary, why? This counts for data manipulation.
14. Did the authors check for the levels of phospho-tau:total tau, data must be provided. This is necessary since the levels of phospho-tau in original blot and the blot provided in the manuscript vary between individual lanes.
55. The authors mention “The concentration of phosphorylated tau and APP in the newborns hippocampus was measured using a colorimetric assay (Bio-rad…). I think this is incorrect since the authors might have measured total protein concentration and not individual proteins. Also, the cat.# of calorimetric assay kit is incorrect.
The English language is adequate, except for very few minor mistakes.
Author Response
Author response to reviewer 2 comments
Reviewer 2 comments
Did the authors perform Y-maze test post treatment with BM-MSCs and GSI-953? If so, the results showing any improvement in the behavior of the animals post-treatment must be provided.
Author response: Thanks for your valuable comment. Y-maze test was carried out for mothers only before pregnancy to ensure memory impairments in mothers.
Old comment: Most importantly, there is missing evidence of the characterization of the BM-MSCs, used in the study, considering they show “fibroblast like morphology”.
Author response: Thank you for your comment. The evidence for the characterization of the BM-MSCs was included in our previous publication (https://www.mdpi.com/1424-8247/16/1/34).
New comment: Although the authors have characterized the BM-MSCs in their earlier studies, the characteristics of the cells may differ from batch to batch and the isolation conditions; therefore, the authors must have performed some analysis on the activity or immunohistochemical analysis of the cells used in this particular study which is missing. In addition, the above-mentioned article must be cited in the method section.
Author response: Thank you for your comment. publication (https://www.mdpi.com/1424-8247/16/1/34) was cited in the method section. The isolation and culture of BM-MSCs was done in the same lab under the same conditions using the same batch of the same age. (Page 4)
The Western blot in Figure 18a. The phospho-tau bands do not correspond with the original blot provided in the supplementary, why? This counts for data manipulation.
Author response: Thank you for your comment. The mistakes in the supplementary files were corrected. Files were included by errors. The correct files of original immunoblots of p-Tau were included in the attached supplementary files.
Did the authors check for the levels of phospho-tau:total tau, data must be provided. This is necessary since the levels of phospho-tau in original blot and the blot provided in the manuscript vary between individual lanes.
Author response: Thank you for your comment. We measured only level of phosphorylated tau which is important for AD disease. Accumulation of phosphorylated tau is a key pathological feature of Alzheimer's disease (https://doi.org/10.1093/brain/awaa223). The correct files of original immunoblots of p-Tau were included in the attached supplementary files.
The authors mention “The concentration of phosphorylated tau and APP in the newborns hippocampus was measured using a colorimetric assay (Bio-rad…). I think this is incorrect since the authors might have measured total protein concentration and not individual proteins. Also, the cat.# of calorimetric assay kit is incorrect.
Author response: Thank you for your comment. Ok, the mistake was corrected and was marked in red colour (Page 7).

Reviewer 3 Report (New Reviewer)
In this manuscript, the authors investigate the protective effects of γ-secretase inhibitor-953 (GSI-953) and bone marrow-derived mesenchymal stem cells (BM-MSCs) using a pregnancy rat model. Via behavioral, histological (cellular), and molecular (ELISA, qRT-PCR, and Western blot) approaches, the authors found that offspring of Aβ25-35-injected dams treated with BM-MSCs or GSI-953 showed significantly improved therapy effect, such as reduced proinflammatory cytokine levels in the serum and reduced number of activated microglia, reduced tau phosphorylation and amyloid precursor protein levels, restored aberrant gene expression in the neonates' hippocampus. Overall, this manuscript provided a panel of meaningful findings elucidating the beneficial effect of GSI-953 and BM-MSCs against AD. Meanwhile, there are several minor concerns that might weaken the manuscript:
1. The title is too long, like a summary of results. Please simplify the title to highlight the most striking conclusion.
2. In the methods section 2.2 Surgical Procedure, it mentions "Rats 140 were subjected to the Y-maze and novel object recognition tests a week after surgery to 141 assess their working and reference memories, respectively." Meanwhile in Figure 1 (experimental design), it mentions the female rats were allowed to be recovered for 10 days and then subjected to behavioral testes. Please clarify a week or 10 days after surgery for behavioral tests.
3. Table 2. Please specify which statistical test method was applied and provide the exact p value for each comparison.
4. From figure 3 to figure 13, is there any statistical test in the comparison of morphological changes between control and treatment groups. If yes, please provide the summary figure (such as a bar plot) and the statistical test information. If not, please mention the limitation of the conclusions drawn from these figures.
Overall, the quality of English language is good.
Author Response
Author response to reviewer 3 comments
Reviewer 3 comments
- The title is too long, like a summary of results. Please simplify the title to highlight the most striking conclusion.
Author response: Thanks for your comment. The title was modified to be: Mesenchymal Stem Cells and Begacestat Mitigate Amyloid-β 25–35-Induced Cognitive Decline in Rat Dams and Hippocampal Deteriorations in Offspring
- In the methods section 2.2 Surgical Procedure, it mentions "Rats were subjected to the Y-maze and novel object recognition tests a week after surgery to assess their working and reference memories, respectively." Meanwhile in Figure 1 (experimental design), it mentions the female rats were allowed to be recovered for 10 days and then subjected to behavioral testes. Please clarify a week or 10 days after surgery for behavioral tests.
Author response: Thanks for your comment. The mistake in surgical procedure was corrected as follow: rats were subjected to the Y-maze and novel object recognition tests after 10 days of surgery to assess their working and reference memories, respectively.
- Table 2. Please specify which statistical test method was applied and provide the exact p value for each comparison.
Author response: Thanks for your comment. The statistical analysis used for behavioral tests was T-test and this was mentioned in the title of Table 2 also and in section “statistical analysis” in Materials and Methods. The F-value was added in table 2 while p value was indicated in table caption.
- From figure 3 to figure 13, is there any statistical test in the comparison of morphological changes between control and treatment groups. If yes, please provide the summary figure (such as a bar plot) and the statistical test information. If not, please mention the limitation of the conclusions drawn from these figures.
Author response: Thanks for your comment. Image J was used to quantify the number, size and dimension of microglial cells of figure 13. The data and statistical analysis of these results were found in Figure 14.

Round 2
Reviewer 2 Report (Previous Reviewer 2)
The authors have addressed the issues raised by this reviewer.
I have one small comment. The authors must update the blots in the section "original blot.pdf".
This manuscript is a resubmission of an earlier submission. The following is a list of the peer review reports and author responses from that submission.
Round 1
Reviewer 1 Report
Mahmoud et al. has submitted their manuscript that is entitled “Treatment with Mesenchymal Stem Cell or γ-Secretase Inhibitor mitigates Amyloid- β 25–35-induced Cognitive Impairment 3 in Dams and Hippocampal Degeneration in Offspring by Rereducing Microglial Activation and Modulating NF-κB and 5 BDNF Signaling” for publication. The author used bone marrow derived mesenchymal stem cells (BM-MSC) or gamma secretase inhibitor (GSI-953) to prevent the single amyloid protein 25-35 injection rat dam model. The author claimed that both MSCs and GSI-953 could repair histopathological changes, inhibit microglial cell activity, improve behavioral impairments, reduce neuroinflammatory cytokines, and decrease protein concentration of P-tau and amyloid precursor protein by increasing activity of brain derived neurotrophic factor and decreasing expression of NF-κB. Thus, the manuscript concluded that the study proved a possible protection against Alzheimer's disease by using mesenchymal stem cells and gamma secretase inhibitors. Overall, the research topic is very interesting and very applicable translational research since the mesenchymal stem cells as therapy for neurological diseases has been widely reported, and the treatment benefits are well accepted. The experimental design and behavior study are very professional, all results are very clear. Data analysis and presentation are faithfully abided to the research results. However, there some major and minor concerns should be addressed before it can be considered for publication:
Major concerns:
1. What is the rationale of aging Aβ25-35 peptide for four days? Is there any data to support the hypothesis?
2. Can the author provide the mechanism on how the injected peptide affects the offspring?
3. What is the distribution of the injected peptide in the brain?
4. It would be very helpful if the author can add cell toxicity result of the aged peptide versus the non-aged peptide to demonstrate the injected peptide is toxic.
5. Please provide more detail information why the MSC has the similar function to the GSI-953
6. The author should provide immunostaining results the any results for amyloid beta of brain tissues from the mother and offsprings.
Minor concerns:
1. Dams were treated 39 with BM-MSCs or GSI-953 against Aβ25-35, showed dramatically reduced in the number and size of activated microglial cells, enhanced the dendritic length, and the elevation of proinflammatory cytokine levels in the offspring were reversed.
2. Lane 82: that lowers Aβ plasma levels in a phase I clinical trail, should be changed to: that lowers plasma Aβ levels in a phase I clinical trail
3. Lane 86: the formation of Aβ1-40 in the brain, cerebrospinal fluid (CSF), and plasma [18]. ].
4. Method section lane 121-122: Free Aβ25–35 was dis- 121 solved to a concentration of 1 mg/mL in 0.9% saline and incubated at 37°C for four days. Please justify the reason for four days incubation
5. Line 133: progressively inject 10 µL Aβ25–35. Please add 10 µL Aβ25–35 (10ug)
6. In 2.3 section: Animal group is very confusion, please rewrite it and indicate the number of each group.
7. Please add animal number each group in table 2
8. Please provide the number of animals of each group to all figures in figure legend
9. Lane 527-530, the author explained the mechanism of GSI-953 by reducing the plaque formation, please explain how a gamma inhibitor has such function. If it is related to amyloid production, has the author tested the brain or plasma Aβ (40 or 42
Reviewer 2 Report
Comments to authors:
In the article “Treatment with Mesenchymal Stem Cell or γ-Secretase Inhibitor mitigates Amyloid- β 25–35-induced Cognitive Impairment in Dams and Hippocampal Degeneration in Offspring by Reducing Microglial Activation and Modulating NF-κB and BDNF Signaling”, the authors investigate the effect of bone-marrow derived mesenchymal stem cells and gamma secretase inhibitor “Begacestat” in mitigating the effect of Amyloid- β 25–35 in neonates of rats post exposure to the fragment. Unfortunately, the study is loosely designed and documented. There are lots of fundamental errors and discrepancies in the work. First of all, I wonder how the authors can identify and differentiate “Glial cells” using simple “Hematoxylin and Eosin staining”. Secondly, the authors mention “dendrites” in “Microglia”, which is incorrect, they are microglial processes, and only neurons have dendrites. In addition, most microglia show ramified to phagocytic morphology in all cases; therefore, neither BM-MSCs nor GSI-953 had any effect in the rodents. Based on the western blot analysis, DMEM alone had equal effect than the potential treatment using BM-MSCs or GSI-953. The statistical analysis is oddly presented. The data from western blot and statistical analysis doesn’t correspond to each other. The magnitude of proposed hypothesis in figure 13 is over-whelming considering all these fundamental flaws.
I also have other major comments:
The authors mentions “The first day of pregnancy was determined by looking for sperm in a vaginal smear.” This doesn’t account for the actual pregnancy. How many successful pregnant females were obtained? What was the average amount of pubs/group?
What does a,b,c,d,e,f mean in the bar graphs?
How was the ELISA analysis performed? Missing details.
How were the size/dimensions of microglia quantified?
Most importantly, there is missing evidence of the characterization of the BM-MSCs, used in the study, considering they show “fibroblast like morphology”.
